# COUNTERFACTUAL GENERATIVE MODELS FOR TIME-VARYING TREATMENTS

## ABSTRACT

Estimating the counterfactual outcome of treatment is essential for decision-making in public health and clinical science, among others. Often, treatments are administered in a sequential, time-varying manner, leading to an exponentially increased number of possible counterfactual outcomes. Furthermore, in modern applications, the outcomes are high-dimensional and conventional average treatment effect estimation fails to capture disparities in individuals. To tackle these challenges, we propose a novel conditional generative framework capable of producing counterfactual samples under time-varying treatment, without the need for explicit density estimation. Our method carefully addresses the distribution mismatch between the observed and counterfactual distributions via a loss function based on inverse probability weighting. We present a thorough evaluation of our method using both synthetic and real-world data. Our results demonstrate that our method is capable of generating high-quality counterfactual samples and outperforms the state-of-the-art baselines.

## 1 INTRODUCTION

Estimating time-varying treatment effect from observational data has garnered significant attention due to the growing prevalence of time-series records. One particular relevant field is public health (Kleinberg & Hripcsak, 2011; Zhang et al., 2017; Bonvini et al., 2021), where researchers and policymakers grapple with a series of decisions on preemptive measures to control epidemic outbreaks, ranging from mask mandates to shutdowns. It is vital to provide accurate and comprehensive outcome estimates under such diverse time-varying treatments, so that policymakers and researchers can accumulate sufficient knowledge and make well-informed decisions with discretion.

In the literature, average treatment effect estimation has received extensive attention and various methods have been proposed (Rosenbaum & Rubin, 1983; Hirano et al., 2003; Imbens, 2004; Lim et al., 2018a; Bica et al., 2020; Berrevoets et al., 2021; Seedat et al., 2022; Melnychuk et al., 2022; Frauen et al., 2023; Vanderschueren et al., 2023). By estimating the average outcome over a population that receives a treatment or policy of interest, these methods evaluate the effectiveness of the treatment via hypothesis testing. However, solely relying on the average treatment effect might not capture the full picture, as it may overlook pronounced disparities in the individual outcomes of the population, especially when the counterfactual distribution is heterogeneous.

Recent efforts (Kim et al., 2018; Kennedy et al., 2023; Melnychuk et al., 2023) have been made to directly estimate the counterfactual density function of the outcome. This idea has demonstrated appealing performance for univariate outcomes. Nonetheless, for multi-dimensional outcomes, the estimation accuracy quickly degrades (Scott & Thompson, 1983). In modern high-dimensional applications, for example, predicting COVID-19 cases at the county level of a state, these methods are hardly scalable and incur a computational overhead.

Adding another layer of complexity, considering time-varying treatments causes the capacity of the potential treatment sequences to expand exponentially. For example, even if the treatment is binary at a single time step, the total number of different combinations on a time-varying treatment increases as $2^d$ with $d$ being the length of history. More importantly, time-varying treatments lead to significant distributional discrepancy between the observed and counterfactual outcomes, as shown in Figure 1.

In this paper, we provide a whole package of accurately estimating high-dimensional counterfactual distributions for time-varying treatments. Instead of a direct density estimation, we implicitly learn

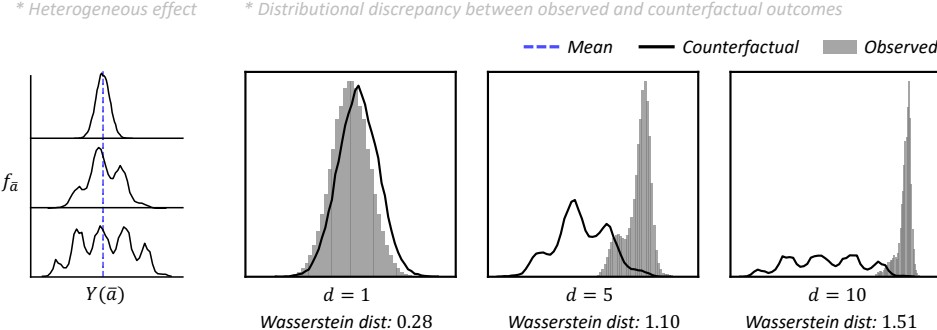

Figure 1: An illustrative example of the counterfactual distribution of a 1-dimensional outcome variable given a time-varying treatment, with different lengths of treatment history. The panel on the left shows that the mean is incapable of describing the heterogeneous effect in counterfactual distributions. The rest of the three panels demonstrate how the length of the treatment history affects the distribution. Here $d$ denotes the length of the history dependence. The longer the dependence on the treatment history, the greater the distributional mismatch tends to be.

the counterfactual distribution by training a generative model, capable of generating credible samples of the counterfactual outcomes given a time-varying treatment. This allows policymakers to assess a policy's efficacy by exploring a range of probable outcomes and deepening their understanding of its counterfactual result. Here, we summarize the benefits of our proposed method:

1. Our model is capable of handling high-dimensional outcomes;

2. Our model outperforms existing state-of-the-art baselines in terms of estimation accuracy and generating high-quality counterfactual samples;

3. Our model enables fast downstream inference, such as average treatment effect estimation and uncertainty quantification.

To be specific, we develop a conditional generator (Mirza & Osindero, 2014; Sohn et al., 2015). This generator, which we choose in a flexible manner, takes into account the treatment history as input and generates counterfactual outcomes that align with the underlying distribution of counterfactuals. The key idea behind the scenes is to utilize a "proxy" conditional distribution as an approximation of the true counterfactual distribution. To achieve this, we establish a statistical relationship between the observed and counterfactual distributions inspired by the g-formula (Neyman, 1923; Rubin, 1978; Robins, 1999; Fitzmaurice et al., 2008). We learn the conditional generator by optimizing a novel weighted loss function based on a pseudo population through Inverse Probability of Treatment Weighting (IPTW) (Robins, 1999). We evaluate our framework through numerical experiments extensively on both synthetic and real-world data sets.

## 1.1 RELATED WORK

Our work has connections to causal inference in time series, counterfactual density estimation, and generative models. To our best knowledge, our work is the first to intersect the three aforementioned areas. Below we review each of these areas independently.

*Causal inference with time-varying treatments.* Causal inference has historically been related to longitudinal data. Classic approaches to analyzing time-varying treatment effects include the g-computation formula, structural nested models, and marginal structural models (Rubin, 1978; Robins, 1986; 1994; Robins et al., 1994; 2000; Fitzmaurice et al., 2008; Li et al., 2021). These seminal works are typically based on parametric models with limited flexibility. Recent advancements in machine learning have significantly accelerated progress in this area using flexible statistical models (Schulam & Saria, 2017; Chen et al., 2023) and deep neural networks (Lim et al., 2018a; Bica et al., 2020; Berrevoets et al., 2021; Li et al., 2021; Seedat et al., 2022; Melnychuk et al., 2022; Frauen et al., 2023; Vanderschueren et al., 2023) to capture the complex temporal dependency of the outcome on treatment and covariate history. These approaches, however, focus on predicting the mean counterfactual outcome instead of the distribution. The performance of these methods also heavily relies on the specific structures (*e.g.*, LSTMs) without more flexible architectures.

*Counterfactual distribution estimation.* Recently, several approaches have emerged to estimate the entire counterfactual distribution rather than the means, including estimating quantiles of the

cumulative distributional functions (CDFs) (Chernozhukov et al., 2013; Wang et al., 2018), re-weighted kernel estimations (DiNardo et al., 1996), and semiparametric methods (Kennedy et al., 2023). In particular, Kennedy et al. (2023) highlights the extra information afforded by estimating the entire counterfactual distribution and using the distance between counterfactual densities as a measure of causal effects. Melnychuk et al. (2023) uses normalizing flow to estimate the interventional density. However, these methods are designed to work under static settings with no time-varying treatments (Alaa & Van Der Schaar, 2017), and are explicit density estimation methods that may be difficult to scale to high-dimensional outcomes. Li et al. (2021) proposes a deep framework based on G-computation which can be used to simulate outcome trajectories on which one can estimate the counterfactual distribution. However, this framework approximates the distribution via empirical estimation of the sample variance, which may be unable to capture the complex variability of the (potentially high-dimensional) distributions. Our work, on the other hand, approximates the counterfactual distribution with a generative model without explicitly estimating its density. This will enable a wider range of application scenarios including continuous treatments and can accommodate more intricate data structures in the high-dimensional outcome settings.

*Counterfactual generative model.* Generative models, including a variety of deep network architectures such as generative adversarial networks (GAN) and autoencoders, have been recently developed to perform counterfactual prediction (Goudet et al., 2017; Louizos et al., 2017; Yoon et al., 2018; Saini et al., 2019; Sauer & Geiger, 2021; Van Looveren et al., 2021; Im et al., 2021; Kuzmanovic et al., 2021; Balazadeh Meresht et al., 2022; Fujii et al., 2022; Liu et al., 2022; Reynaud et al., 2022; Zhang et al., 2022). However, many of these approaches primarily focus on using representation learning to improve treatment effect estimation rather than obtaining counterfactual samples or approximating counterfactual distributions. For example, Yoon et al. (2018); Saini et al. (2019) adopt deep generative models to improve the estimation of individual treatment effects (ITEs) under static settings. Some of these approaches focus on exploring causal relationships between components of an image (Sauer & Geiger, 2021; Van Looveren et al., 2021; Reynaud et al., 2022). Furthermore, there has been limited exploration of applying generative models to time series settings in the existing literature. A few attempts, including Louizos et al. (2017); Kuzmanovic et al. (2021), train autoencoders to estimate treatment effect using longitudinal data. Nevertheless, these methods are not intended for drawing counterfactual samples. In sum, to the best of our knowledge, our work is the first to use generative models to approximate counterfactual distribution from data with time-varying treatments, a novel setting not addressed by prior works.

## 2 METHODOLOGY

### 2.1 PROBLEM SETUP

In this study, we consider the treatment for each discrete time period (such as day or week) as a random variable $A_t \in \mathcal{A} = \{0, 1\}$, where $t = 1, \ldots, T$ and $T$ is the total number of time points. Note that our framework also works with categorical and even continuous treatments. Let $X_t \in \mathcal{X} \subset \mathbb{R}^h$ be the time-varying covariates, and $Y_t \in \mathcal{Y} \subset \mathbb{R}^m$ the subject's outcome at time $t$. We use $\overline{A}_t = \{A_{t-d+1}, \ldots, A_t\}$ to denote the previous treatment history from time $t - d + 1$ to $t$, where $d$ is the length of history dependence. Similarly, we use $\overline{X}_t = \{X_{t-d+1}, \ldots, X_t\}$ to denote the covariate history. We use $y_t$, $a_t$, and $x_t$ to represent a realization of $Y_t$, $A_t$, and $X_t$, respectively, and use $\overline{a}_t = (a_{t-d+1}, \ldots, a_t)$ and $\overline{x}_t = (x_{t-d+1}, \ldots, x_t)$ to denote the history of treatment and covariate realizations. In the sections below, we will refer to $Y_t$, $\overline{A}_t$, and $\overline{X}_t$ as simply $Y$, $\overline{A}$, and $\overline{X}$, where $t$ will be clear from context. Since the outcome is independent conditioning on its history, we can consider the samples across time and individuals as conditionally independent tuples $(y^i, \overline{a}^i, \overline{x}^i)$, where $i$ denotes the sample index and the sample size is $N$.

The goal of our study is to obtain realistic samples of the counterfactual outcome for all given time-varying treatment $\overline{a}$, without estimating its counterfactual density. Let $Y(\overline{a})$ denote the counterfactual outcome for a subject under a time-varying treatment $\overline{a}$, and define $f_{\overline{a}}$ as its counterfactual distribution. We note that $f_{\overline{a}}$ is different from the marginal density of $Y$, as the treatment is fixed at $\overline{A} = \overline{a}$ in the conditioning. It is also not equal to the unadjusted conditional density $f(y|\overline{a})$. Instead, $f_{\overline{a}}$ is the density of the counterfactual variable $Y(\overline{a})$, which represents the outcome that would have been observed if treatment were set to $\overline{A} = \overline{a}$. We assume the standard assumptions: consistency, positivity and sequential strong ignorability are hold (Robins et al., 2000; Lim et al., 2018a). See Appendix A for more details about these assumptions. We also assume that $Y$, $\overline{A}$, and $\overline{X}$ follows the typically

structural causal relationship as shown in Figure 8 (Appendix B), which is a classical setting in longitudinal causal inference (Robins, 1986; Robins et al., 2000).

## 2.2 COUNTERFACTUAL GENERATIVE FRAMEWORK FOR TIME-VARYING TREATMENTS

This paper proposes a counterfactual generator, denoted as $g_\theta$, to simulate $Y(\overline{a})$ according to the *proxy conditional distribution* $f_\theta(y|\overline{a})$ instead of directly modeling its expectation or specifying a parametric counterfactual distribution. Here we use $\theta \in \Theta$ to represent the model's parameters, and formally define the generator as a function:

$$g_\theta(z, \overline{a}) : \mathbb{R}^r \times \mathcal{A}^d \to \mathcal{Y}. \tag{1}$$

The generator takes as input a random noise vector ($z \in \mathbb{R}^r \sim \mathcal{N}(0, I)$) and the time-varying treatment $\overline{a}$. The output of the generator is a sample of possible counterfactual outcomes that follows the proxy conditional distribution represented by $\theta$, *i.e.*,

$$y \sim f_\theta(\cdot|\overline{a}),$$

which can be viewed as an approximate of the underlying counterfactual distribution $f_{\overline{a}}$. Figure 2 shows an overview of the proposed generative model architecture.

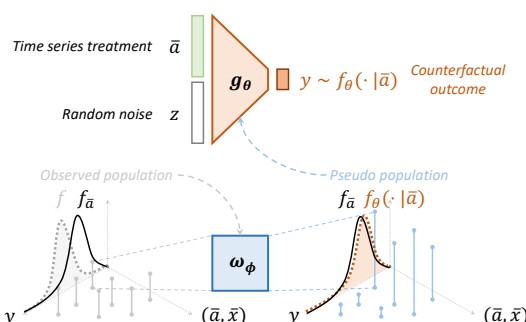

Figure 2: The architecture of the proposed counterfactual generative models. The generator $g_\theta$ is designed to produce samples of the outcome variable $Y(\overline{a})$ with a given time-varying treatment $\overline{a}$. The generated samples are expected to conform to the proxy conditional distribution $f_\theta$, which is an approximate of the underlying counterfactual distribution $f_{\overline{a}}$.

**Marginal structural generative models** The learning objective is to find the optimal generator that minimizes the distance between the proxy conditional distribution $f_\theta(\cdot|\overline{a})$ and the true counterfactual distribution $f_{\overline{a}}$ for any treatment sequence, $\overline{a}$, as illustrated in Figure 3. For a general distributional distance, $D_f(\cdot, \cdot)$, the objective can be expressed as finding an optimal $\hat{\theta}$ that minimizes the difference between $f_\theta(\cdot|\overline{a})$ and $f_{\overline{a}}$ over all treatment $\overline{a}$:

$$\hat{\theta} = \arg\min_{\theta \in \Theta} \mathbb{E}_{\overline{a}} \left[ D_f(f_{\overline{a}}, f_\theta(\cdot|\overline{a})) \right]. \tag{2}$$

If the distance metric is Kullback-Leibler (KL) divergence, this objective can be expressed equivalently by maximizing the log-likelihood (Murphy (2012), Proof in Appendix C) :

$$\hat{\theta} = \arg\min_{\theta \in \Theta} \mathbb{E}_{\overline{a}} \left[ \mathbb{E}_{y \sim f_{\overline{a}}} \log f_\theta(\cdot|\overline{a}) \right]. \tag{3}$$

To obtain samples from the counterfactual distribution $f_{\overline{a}}$, we follow the idea of marginal structural models (MSMs) introduced by Neyman (1923); Rubin (1978); Robins (1999) and extended by Fitzmaurice et al. (2008) to account for time-varying treatments. Specifically, we introduce Lemma 1, which follows the g-formula proposed in Robins (1999) and establishes a connection between the counterfactual distribution and the data distribution. The proof can be found in Appendix B.

**Lemma 1.** *Under unconfoundedness and positivity, we have*

$$f_{\overline{a}}(y) = \int \frac{1}{\prod_{\tau=t-d+1}^{t} f\left(a_\tau | \overline{a}_{\tau-1}, \overline{X}_\tau\right)} f\left(y, \overline{a}, \overline{X}\right) d\overline{X} \tag{4}$$

*where $f$ denotes the observed data distribution.*

Now we present a proposition using Lemma 1, allowing us to substitute the expectation in (3), computed over a counterfactual distribution, with the sample average over a pseudo-population. This pseudo-population is constructed by assigning weights to each data tuple based on their subject-specific IPTW. Figure 3 gives an illustration of the learning objective. See the proof in Appendix D.

**Proposition 1.** *Let $\mathcal{D}$ denote the set of observed data tuples. The generative learning objective can be approximated by:*

$$\mathbb{E}_{\overline{a}}\left[\mathbb{E}_{y \sim f_{\overline{a}}} \log f_\theta(y|\overline{a})\right] \approx \frac{1}{N} \sum_{(y,\overline{a},\overline{x}) \in \mathcal{D}} w_\phi(\overline{a}, \overline{x}) \log f_\theta(y|\overline{a}), \tag{5}$$

*where $N$ represents the sample size, and $w_\phi(\overline{a}, \overline{x})$ denotes the subject-specific IPTW, parameterized by $\phi \in \Phi$, which takes the form:*

$$w_\phi(\overline{a}, \overline{x}) = \frac{1}{\prod_{\tau=t-d+1}^{t} f_\phi(a_\tau | \overline{a}_{\tau-1}, \overline{x}_\tau)}. \tag{6}$$

**Remark 1.** *The generative learning objective in Proposition 1 offers specific benefits when compared to plugin methods using Lemma 1 (Bickel & Kwon, 2001; Kim et al., 2018) and density estimators (Melnychuk et al., 2023). The use of IPTW instead of a doubly robust method is due to the practical challenges in combining IPTW with an outcome based model for approximating the high-dimensional counterfactual distribution under time-varying confounders. We include a detailed discussion in Appendix E.*

Here we use another model, denoted by $\phi \in \Phi$, to represent the conditional probability $f(A_\tau | \overline{A}_\tau, \overline{X}_\tau)$, which defines the IPTW $w_\phi$ and can be learned separately using observed data. Note that the effectiveness of this method is dependent on a correct specification of the IPTW $\phi$, *i.e.*, the black dot is inside of the blue area in Figure 3 (Fitzmaurice et al., 2008; Lim et al., 2018a). In Lim et al. (2018a), they use an RNN-like structure to represent the conditional probability $f_\phi$ without making strong assumptions on the form of the conditional probability. The choice of $g_\theta$ and $f_\phi$ are entirely general and both can be specified using deep architectures. In this paper, we use fully-connected neural networks for both $g_\theta$ and $f_\phi$.

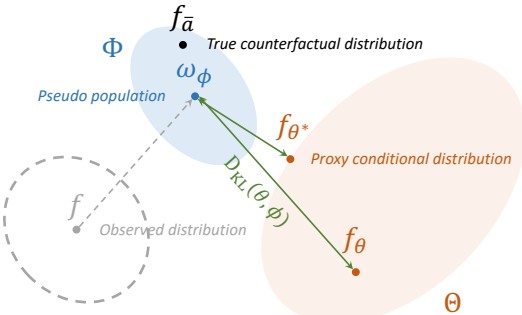

Figure 3: An illustration of our learning objective. We aim to minimize the KL-divergence between the proxy conditional distribution $f_\theta(\cdot|\overline{a})$ and the true counterfactual distribution $f_{\overline{a}}$. Similarly, if we use the Wasserstein

**Variational approximation and learning** To compute the weighted log-likelihood as expressed in (5) and learn the proposed generative model, we can leverage various state-of-the-art generative learning algorithms, such as conditional normalizing flow (Bhattacharyya et al., 2019) and guided diffusion models (Dhariwal & Nichol, 2021). In this paper, we adopt the conditional variational autoencoder (CVAE) (Sohn et al., 2015), a commonly-used learning algorithm for generative models, approximate the logarithm of the proxy conditional probability using its evidence lower bound (ELBO):

$$\log f_\theta(y|\overline{a}) \geq -D_{\text{KL}}\left(q(z|y,\overline{a})||p_\theta(z|\overline{a})\right) + \mathbb{E}_{q(z|y,\overline{a})}\left[\log p_\theta(y|z,\overline{a})\right], \tag{7}$$

---

**Algorithm 1** Learning algorithm for the conditional generator $\theta$

---

**Input:** Training set $\mathcal{D}$ data tuples: $\mathcal{D} = \{(y^i, \overline{a}^i, \overline{x}^i)\}_{i=1,\dots,N}$ where $N$ is the total number of training samples; the number of the learning epochs $E$.

**Initialization:** model parameters $\theta$ and fitted $\widehat{\phi}$ using $\mathcal{D}$.

**while** $e < E$ **do**

    **for** each sampled batch $\mathcal{B}$ with size $n$ **do**

        1. Draw samples $\epsilon \sim \mathcal{N}(0, I)$ from noise distribution, where $I$ is the identity matrix;

        2. Compute the ELBO of $\log f_\theta(y|\overline{a})$ for $(y, \overline{a}, \overline{x}) \in \mathcal{B}$ given $\epsilon$ and $\theta$ according to (7);

        3. Re-weight the ELBO for $(y, \overline{a}, \overline{x}) \in \mathcal{B}$ using $w_{\widehat{\phi}}(\overline{a}, \overline{x})$ according to (9);

        4. Update $\theta$ using stochastic gradient descent by maximizing (5).

    **end for**

**end while**

**return** $\theta$

---

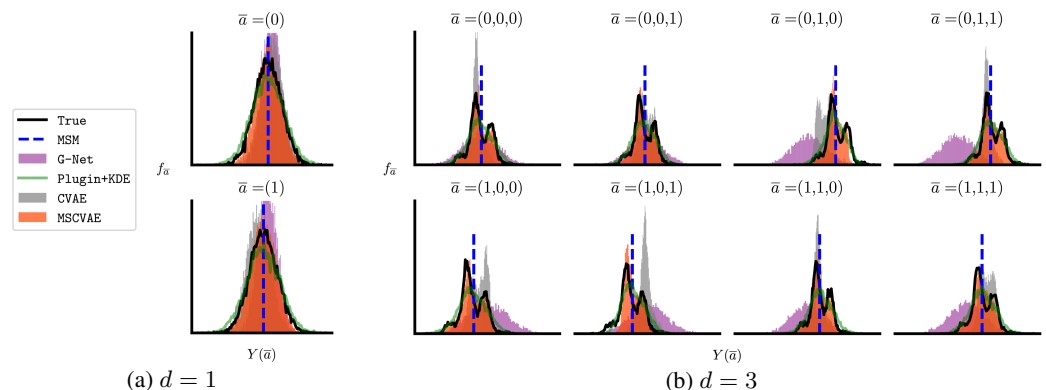

Figure 4: The estimated and true counterfactual distributions across various lengths of history dependence ($d = 1, 3$) on the fully synthetic datasets ($m = 1$). Each sub-panel provides a comparison for a specific treatment combination $\overline{a}$. We include the plot for $d = 5$ in Appendix H.

Table 1: Quantitative performance on fully-synthetic and semi-synthetic data

| | Fully synthetic ($m = 1$) | | | | | | COVID-19 | TV-MNIST |
| | $d = 1$ | | $d = 3$ | | $d = 5$ | | $m = 67$ | $m = 784$ |
| Methods | Mean ↓ | Wasserstein ↓ | Mean ↓ | Wasserstein ↓ | Mean ↓ | Wasserstein ↓ | FID* ↓ | FID* ↓ |
|---|---|---|---|---|---|---|---|---|
| MSM+NN | **0.001** (**0.002**) | 0.601 (0.603) | 0.070 (**0.159**) | 0.689 (0.718) | 0.198 (**0.563**) | 0.600 (0.737) | 1.085 (1.665) | **1.236** (3.956) |
| KDE | 0.246 (0.267) | 0.244 (0.268) | 0.520 (1.080) | 0.538 (1.080) | 0.538 (1.419) | 0.539 (1.419) | 0.981 (2.665) | 1.509 (2.557) |
| Plugin+KDE | 0.010 (0.014) | **0.034** (**0.036**) | **0.045** (0.168) | **0.132** (**0.168**) | **0.147** (0.598) | **0.182** (**0.598**) | 0.652 (**0.759**) | 1.370 (**1.799**) |
| G-Net | 0.211 (0.258) | 0.572 (0.582) | 1.167 (2.173) | 1.284 (2.173) | 2.314 (5.263) | 2.354 (5.263) | 0.965 (1.856) | 1.751 (6.096) |
| CVAE | 0.250 (0.287) | 0.253 (0.288) | 0.517 (1.061) | 0.553 (1.061) | 0.539 (1.430) | 0.613 (1.430) | **0.641** (2.654) | 2.149 (5.484) |
| MSCVAE | **0.006** (**0.006**) | **0.055** (**0.056**) | **0.046** (**0.150**) | 0.105 (**0.216**) | **0.150** (**0.633**) | **0.173** (**0.633**) | **0.336** (**0.712**) | **0.270** (**1.004**) |

* Numbers represent the average metric across all treatment combinations and those in the parentheses represent the worst across treatment combinations.

where $q$ is a variational approximation of the posterior distribution over the random noise given observed outcome $y$ and its treatment $\overline{a}$. The first term on the right-hand side is the Kullback–Leibler (KL) divergence of the approximate posterior $q(\cdot|y, \overline{a})$ from the exact posterior $p_\theta(\cdot|\overline{a})$. The second term is the log-likelihood of the latent data-generating process. The complete derivation of (7) and implementation details can be found in Appendix F. We summarize our learning procedure in Algorithm 1.

## 3 EXPERIMENTS

We evaluate our method using numerical examples and demonstrate the superior performance compared to five state-of-the-art methods. These are (1) Kernel Density Estimation (KDE) (Rosenblatt, 1956), (2) Marginal structural model with a fully-connected neural network (MSM+NN) (Robins et al., 1999; Lim et al., 2018a), (3) Conditional Variational Autoencoder (CVAE) (Sohn et al., 2015), (4) Semi-parametric Plug-in method based on pseudo-population (Plugin+KDE) (Kim et al., 2018), and (5) G-Net (G-Net) (Li et al., 2021). In the following, we refer to our proposed conditional event generator as marginal structural conditional variational autoencoder (MSCVAE). See Appendix G.1 for a detailed review of these baseline methods. Here, the G-Net is based on G-computation. The Plugin+KDE is tailored for counterfactual density estimation. The CVAE acts as a reference model, highlighting the significance of IPTW reweighting in our approach.

**Experiment set-up** To learn the model parameter $\theta$, we use stochastic gradient descent to maximize the weighted log-likelihood (5). We adopt an Adam optimizer (Kingma & Ba, 2014) with a learning rate of $10^{-3}$ and a batch size of 256. To ensure learning stability, we follow a commonly-used practice (Xiao et al., 2010; Lim et al., 2018a) that involves truncating the subject-specific IPTW weights at the 0.01-th and 99.99-th percentiles and normalizing them by their mean. All experiments are performed on Jupyter Notebook with 16GB RAM and a 2.6 GHz 6-Core Intel Core i7 CPU. More details of the experiment set-up can be found in Appendix G.3.

### 3.1 FULLY SYNTHETIC DATA

We first assess the effectiveness of the MSCVAE using fully synthetic experiments. Following the classical experimental setting described in Robins et al. (1999), we simulate three synthetic datasets with different lengths of history dependence ($d = 1, 3, 5$) using linear models. Each dataset

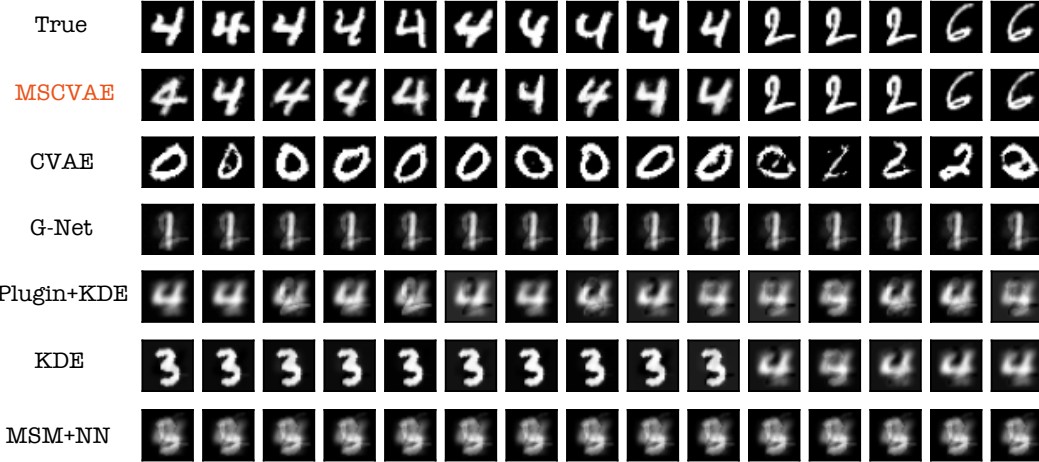

Figure 5: Results on the semi-sythetic TV-MNIST datasets ($m = 784$). We show representative samples generated from different methods under the treatment combinations $\bar{a} = (1, 1, 1)$.

comprises $10,000$ trajectories, representing recorded observations of individual subjects. These trajectories consist of 100 data tuples, encompassing treatment, covariate, and outcome values at specific time points. See Appendix G.4 for a detailed description of the synthetic data generation. The causal dependence between these variables is visualized in Figure 8 (Appendix B). In Figure 4, the `MSCVAE` (orange shading) outperforms the baseline methods in accurately capturing the shape of the true counterfactual distributions (represented by the black line) across all scenarios. It is worth mentioning that the learned distribution produced by `CVAE` deviates significantly from the desired target, emphasizing the significance of the weighting term in Proposition 1 in accurately approximating the counterfactual distribution.

Table 1 summarizes the quantitative comparisons across the baselines. For the fully synthetic datasets, we adopt two metrics: mean distance and 1-Wasserstein distance (Frogner et al., 2015; Panaretos & Zemel, 2019), as commonly-used metrics to measure the discrepancies between the approximated and counterfactual distributions (see Appendix G.2 for more details). The `MSCVAE` not only consistently achieves the smallest Wasserstein distance in the majority of the experimental settings, but also demonstrates highly competitive accuracy on mean estimation, which is consistent with the result in Figure 4. Note that even though our goal is not to explicitly estimate the counterfactual distribution, the results clearly demonstrate that our generative model can still accurately approximate the underlying counterfactual distribution, even compared to the unbiased density-based method such as `Plugin+KDE`.

## 3.2 SEMI-SYNTHETIC DATA

To demonstrate the ability of our generative framework to generate credible high-dimensional counterfactual samples, we test our method on two semi-synthetic datasets. The benefit of these datasets is that both factual and counterfactual outcomes are available. Therefore, we can obtain a sample from the ground-truth counterfactual distribution, which we can then use for benchmarking. We evaluate the performance by measuring the quality of generated samples and the true samples from the dataset.

**Time-varying MNIST** We create TV-MNIST, a semi-synthetic dataset using MNIST images (Deng, 2012; Jesson et al., 2021) as the outcome variable ($m = 784$). In this dataset, images are randomly selected, driven by the result of a latent process defined by a linear autoregressive model, which takes a 1-dimensional covariate and treatment variable as inputs and outputs a digit (between 0 and 9). Here we set the length of history dependence, $d$, to 3. This setup allows us to evaluate the performance of the algorithms by visually contrasting the quality and distribution of generated samples against counterfactual ones. The full description of the dataset can be found in Appendix G.5.

**Pennsylvania COVID-19 mask mandate** We create another semi-synthetic dataset to investigate the effectiveness of mask mandates in Pennsylvania during the COVID-19 pandemic. We collected data from multiple sources, including the Centers for Disease Control and Prevention (CDC), the

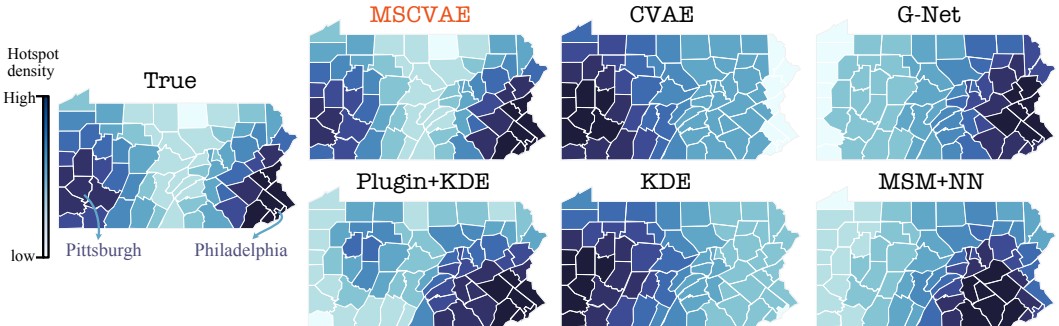

Figure 6: Results on the semi-synthetic Pennsylvania COVID-19 mask datasets ($m = 67$) under the treatment combination $\bar{a} = (1, 1, 1)$. We visualize the distribution of "hotspots" from the generated and true counterfactual distribution. For each model, we generate $500$ counterfactual samples. Each sample is a 67-dimensional vector representing the inferred new cases per 100K for the counties in Pennsylvania. We define the hotspot of each sample as the coordinate of the county with the highest number of new cases per 100K, and visualize the density of the 500 hotspots using kernel density estimation.

US Census Bureau, and a Facebook survey (Zhu et al., 2021; for Disease Control, 2021; Zhu et al., 2022; Google, 2022; Bureau, 2022; Group, 2022). The dataset encompasses variables aggregated on a weekly basis spanning 106 weeks from 2020 and 2022. There are four state-level covariates (per 100K people): the number of deaths, the average retail and recreation mobility, the surveyed COVID-19 symptoms, and the number of administered COVID-19 vaccine doses. We set the state-level mask mandate policy (with values of $0$ indicating no mandate and $1$ indicating a mandate) as the treatment variable, and the county-level number of new COVID-19 cases (per 100K) as the outcome variable ($m = 67$). We simulate $2,000$ trajectories of the (covariate, treatment) tuples of $300$ time points (each point corresponding to a week) according to the real data. The outcome model is structured to exhibit a peak, defined as the "hotspot", in one of the state's two major cities: Pittsburgh or Philadelphia. The likelihood of these hotspots is contingent on the covariates. Consequently, the counterfactual and observed distributions manifest as bimodal, with varying probabilities for the hotspot locations. To ensure a pertinent analysis window, we've fixed the history dependence length, $d$, at 3, aligning with the typical duration within which most COVID-19 symptoms recede (Maltezou et al., 2021). The full description of the dataset can be found in Appendix G.6.

Given the high-dimensional outcomes of both semi-synthetic datasets, straightforward comparisons using means or the Wasserstein distance of the distributions tend to be less insightful. As a result, we use FID* (Fréchet inception distance *), an adaptation of the commonly-used FID (Heusel et al., 2017) to evaluate the quality of the counterfactual samples. For the TV-MNIST dataset, we utilize a pre-trained MNIST classifier, and for the COVID-19 dataset, a geographical projector, to map the samples into a feature space. Subsequently, we calculate the Wasserstein distance between the projected samples and counterfactual samples. The details can be found in Appendix G.2.

As we can observe in Figure 5 and 6, the MSCVAE outperforms other baselines in generating samples that closely resembles the ground truth. This visual superiority is also reinforced by the overwhelmingly better FID* scores of MSCVAE compared to other baselines methods, as shown in Table 1. It's worth noting that the samples produced by the Plugin+KDE appear blurred in Figure 5 and exhibit noise in Figure 6. This can be attributed to the inherent complexities of high-dimensional density estimation (Scott & Thompson, 1983). Such observations underscore the value of employing a generative model to craft high-dimensional samples without resorting to precise density estimation. We also notice that G-Net fails to capture the high-dimensional counterfactual outcomes, particularly due to challenges in accurately defining the conditional outcome model and the covariate density model. The superior results of MSCVAE compared to CVAE and of Plugin+KDE over KDE emphasize the pivotal role of IPTW correction during modeling. Moreover, deterministic approaches like MSM+NN might fall short in capturing key features of the counterfactual distribution. In sum, the semi-synthetic experiments highlights the distinct benefits of our generative framework, particularly in generating high-quality counterfactual samples under time-varying treatments in a high-dimensional causal context.

## 3.3 REAL DATA

We perform a case study using a real COVID-19 mask mandate dataset across the U.S. from 2020 to 2021 spanning 49 weeks. We analyze the same set of variables as the semi-synthetic COVID-19 dataset except that: (1) We exclude the vaccine dosage from the covariates due to missing data in some states. (2) All variables in this dataset, including the treatments, the covariates, and the outcomes, are real data without simulation. Due to the limitation on the sample size for state-level observations, we only look at the county-level data, covering $3,219$ U.S. counties. This leads to $m = 1$. The details can be found in Appendix G.6.

Figure 7 illustrates a comparative analysis of the distribution of the observed and generated outcome samples under two different scenarios: one without a mask mandate ($\overline{a} = (0,0,0)$) and the other with a full mask mandate ($\overline{a} = (1,1,1)$). In the left panel, we observe that the distributions under both policies appear remarkably similar, suggesting that the mask mandate has a limited impact on controlling the spread of the virus. This unexpected outcome challenges commonly held assumptions. In the right panel, we present counterfactual distributions estimated using our method, revealing a noticeable disparity between the mask mandate and no mask mandate scenarios. The mean of the distribution for the mask mandate is significantly lower than that of the no-mask mandate. These

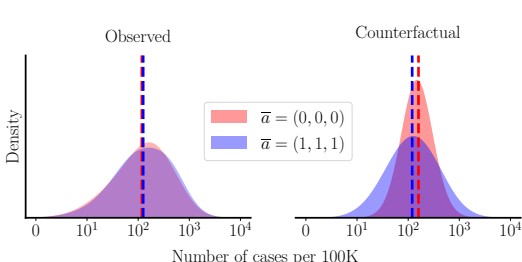

Figure 7: Observed distribution and estimated counterfactual distribution of the number of real COVID-19 cases per 100K under two mask policies. The vertical dashed lines represent the mean of the corresponding distributions.

findings indicate that implementing a mask mandate consistently for three consecutive weeks can effectively reduce the number of future new cases. It aligns with the understanding supported by health experts' suggestions and various studies (Van Dyke et al., 2020; Adjodah et al., 2021; Guy Jr et al., 2021; Nguyen, 2021; Wang et al., 2021) regarding the effectiveness of wearing masks. Finally, it is important to note that the implementation of full mask mandates exhibits a significantly higher variance compared to the absence of a mask mandate. This implies that the impact of a mask mandate varies across different data points, specifically counties in our study. This insight highlights the need for policymakers to carefully assess the unique characteristics of their respective regions when considering the implementation of mask mandate policies. It is crucial for policymakers to understand that the effectiveness of a mask mandate may yield negative outcomes in certain extreme cases. Therefore, when proposing and implementing such policies, a thorough examination of the specific circumstances is highly recommended to avoid any unintended consequences.

## 4 CONCLUSIONS

We have introduced a powerful conditional generative framework tailored to generate samples that mirror counterfactual distributions in scenarios where treatments vary over time. Our model approximates the true counterfactual distribution by minimizing the KL-divergence between the true distribution and a proxy conditional distribution, approximated by generated samples. We have showcased our framework's superior performance against state-of-the-art methods in both fully-synthetic and real experiments.

Our proposed framework has great potential in generating intricate high-dimensional counterfactual outcomes. For example, our model can be enhanced by adopting the cutting edge generative models and their learning algorithms, such as diffusion models. Additionally, our generative approach can be easily adapted to scenarios with continuous treatments, where the conditional generator enables extrapolation between unseen treatments under continuity assumptions.

We also recognize potential caveats stemming from breaches in statistical assumptions. In real-world scenarios, the conditional exchangeability condition might be compromised due to unobserved confounders. Similarly, the positivity assumption could be at risk, attributed to the escalating number of treatment combinations as $d$ increases. Hence, meticulous assessment of these assumptions is imperative for a thorough and accurate statistical interpretation when employing our framework.

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

## A  Assumptions

The standard assumptions needed for identifying the treatment effects are (Fitzmaurice et al., 2008; Lim et al., 2018a; Schulam & Saria, 2017):

1. *Consistency*: If $\overline{A}_t = \overline{a}_t$ for a given subject, then the counterfactual outcome for treatment, $\overline{a}_t$, is the same as the observed (factual) outcome: $Y(\overline{a}_t) = Y$.

2. *Positivity*: If $\mathbb{P}\{\overline{A}_{t-1} = \overline{a}_{t-1}, \overline{X}_t = \overline{x}_t\} \neq 0$, then $\mathbb{P}\{\overline{A}_t = \overline{a}_t | \overline{A}_{t-1} = \overline{a}_{t-1}, \overline{X}_t = \overline{x}_t\} > 0$ for all $\overline{a}_t$ (Imai & Van Dyk, 2004).

3. *Sequential strong ignorability*: $Y(\overline{a}_t) \perp\!\!\!\perp \overline{A}_t | \overline{A}_{t-1} = \overline{a}_{t-1}, \overline{X}_t = \overline{x}_t$, for all $a_t$ and $t$.

Assumption 2 means that, for each timestep, each treatment has a non-zero probability of being assigned. Assumption 3 (also called conditional exchangeability) means that there are no unmeasured confounders, that is, all of the covariates affecting both the treatment assignment and the outcomes are present in the the observational dataset. Note that while assumption 3 is standard across all methods for estimating treatment effects, it is not testable in practice (Pearl, 2009; Robins et al., 2000).

## B  Proof of Lemma 1

Given a probability distribution for $(Y, \overline{A}, \overline{X})$ and a causal directed acyclic graph (DAG) shown in Figure 8, we can factor $f(Y, \overline{A}, \overline{X})$ as

$$f(Y, \overline{A}, \overline{X}) = f\left(Y | \overline{A}, \overline{X}\right) \prod_{\tau=t-d+1}^{t} f\left(X_\tau | \overline{A}_{\tau-1}, \overline{X}_{\tau-1}\right) \prod_{\tau=t-d+1}^{t} f\left(A_\tau | \overline{A}_{\tau-1}, \overline{X}_\tau\right). \quad (8)$$

Using the definition of g-formula (Robins, 1999), we have

$$\begin{aligned}
f_{\overline{a}}(y) &= \int f\left(y | \overline{a}, \overline{X}\right) \cdot \prod_{\tau=t-d+1}^{t} f\left(X_\tau | \overline{a}_{\tau-1}, \overline{X}_{\tau-1}\right) d\overline{X} \\
&= \int f\left(y | \overline{a}, \overline{X}\right) \cdot \frac{\prod_{\tau=t-d+1}^{t} f\left(a_\tau | \overline{a}_{\tau-1}, \overline{X}_\tau\right)}{\prod_{\tau=t-d+1}^{t} f\left(a_\tau | \overline{a}_{\tau-1}, \overline{X}_\tau\right)} \cdot \prod_{\tau=t-d+1}^{t} f\left(X_\tau | \overline{a}_{\tau-1}, \overline{X}_{\tau-1}\right) d\overline{X} \\
&\overset{(i)}{=} \int \frac{1}{\prod_{\tau=t-d+1}^{t} f\left(a_\tau | \overline{a}_{\tau-1}, \overline{X}_\tau\right)} f\left(y, \overline{a}, \overline{X}\right) d\overline{X},
\end{aligned}$$

where the equation $(i)$ holds due to (8).

## C  Proof of Equation 3

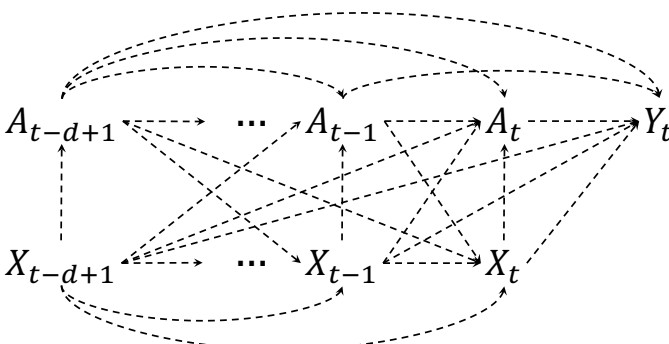

Figure 8: The causal directed acyclic graph (DAG) of the time-varying treatment.

The objective function for training the counterfactual generator minimizes the difference between $f_\theta(\cdot|\overline{a})$ and the true counterfactual distribution $f_{\overline{a}}$ with respect to a distributional difference $D_f(\cdot,\cdot)$ over all treatment combinations, $\overline{a}$. When the distance measure is the KL-divergence, Equation 2 can be written as:

$$
\begin{aligned}
\hat{\theta} &= \underset{\theta\in\Theta}{\arg\min}\ \underset{\overline{a}}{\mathbb{E}}\ [\mathrm{KL}(f_{\overline{a}}(\cdot)||f_\theta(\cdot|\overline{a}))] \\
&= \underset{\theta\in\Theta}{\arg\min}\ \underset{\overline{a}}{\mathbb{E}}\left[\int \log\left(\frac{f_{\overline{a}}(y)}{f_\theta(y|\overline{a})}\right)f_{\overline{a}}(y)dy\right] \\
&= \underset{\theta\in\Theta}{\arg\max}\ \underset{\overline{a}}{\mathbb{E}}\left[\int \log\left(f_\theta(y|\overline{a})\right)f_{\overline{a}}(y)dy\right] \\
&= \underset{\theta\in\Theta}{\arg\max}\ \underset{\overline{a}}{\mathbb{E}}\left[\underset{y\sim f_{\overline{a}}}{\mathbb{E}}\ \log f_\theta(\cdot|\overline{a})\right]
\end{aligned}
$$

## D   Proof of Proposition 1

We recall our notations for densities: $f(y,\overline{a},\overline{x})$ denotes the density of the observed data, $f_{\overline{a}}$ denotes the counterfactual density under $\overline{a}$, and $f_\theta(\cdot|\overline{a})$ denotes the conditional density represented by our conditional generator. Note that these density notations should be interpreted in a broad sense to unify discrete and continuous random variables, meaning that when $\bar{a}$ is a discrete random variable, we allow the density function to be Delta functions. For example, when $\bar{A}$ is distributed as $\mathbb{P}(\bar{A}=\bar{a}_1)=\mathbb{P}(\bar{A}=\bar{a}_2)=0.5$, its corresponding density function is $p(\cdot)=0.5\delta_{\bar{a}_1}(\cdot)+0.5\delta_{\bar{a}_2}(\cdot)$.

We also recall that $w(\overline{a},\overline{x})=1/\prod_{\tau=t-d+1}^{t}f\left(a_\tau|\overline{a}_{\tau-1},\overline{x}_\tau\right)$ where $f\left(a_\tau|\overline{a}_{\tau-1},\overline{x}_\tau\right)$ denotes the individual propensity score. Here $\bar{A}$ is the collection of all the history actions. Using Lemma 1, we have

$$
\begin{aligned}
&\underset{\overline{a}}{\mathbb{E}}\left[\underset{y\sim f_{\overline{a}}}{\mathbb{E}}[\log f_\theta(y|\overline{a})]\right] \\
&= \underset{\overline{a}}{\mathbb{E}}\left[\int \log f_\theta(y|\overline{a})f_{\overline{a}}(y)\,dy\right] \\
&= \underset{\overline{a}}{\mathbb{E}}\left[\int \log f_\theta(y|\overline{a})\int w(\overline{a},\overline{X})f\left(y,\overline{a},\overline{X}\right)d\overline{X}dy\right] \qquad \text{(by Lemma 1)} \\
&= \underset{\overline{a}}{\mathbb{E}}\left[\int\int \log f_\theta(y|\overline{a})w(\overline{a},\overline{X})f\left(y,\overline{a},\overline{X}\right)d\overline{X}dy\right] \\
&= \underset{(y,\overline{a},\overline{x})\sim f}{\mathbb{E}}w(\overline{a},\overline{x})\log f_\theta(y|\overline{a}) \\
&\approx \frac{1}{N}\sum_{(y,\overline{a},\overline{x})\in\mathcal{D}}w_\phi(\overline{a},\overline{x})\log f_\theta(y|\overline{a}), \qquad \text{(Empirical estimation)}
\end{aligned}
$$

where $N$ represents the sample size, and $w_\phi(\overline{a},\overline{x})$ denotes the learned subject-specific IPTW, parameterized by $\phi\in\Phi$, which takes the form:

$$
w_\phi(\overline{a},\overline{x}) = \frac{1}{\prod_{\tau=t-d+1}^{t}f_\phi(a_\tau|\overline{a}_{\tau-1},\overline{x}_\tau)}. \tag{9}
$$

## E   Connection to related methods

**Plug-in density estimation**   Plug-in approaches have been commonly used to estimate the counterfactual density in the static setting(Bickel & Kwon, 2001; Kim et al., 2018; Kennedy et al., 2023) and

can be extended to our time-varying setting via direct application of Lemma 1. However, this practice could be problematic when the sample size is large as it requires averaging the entire observed dataset for each evaluation of $y$. Instead, we circumvent this computational challenge by approximating the counterfactual density using a proxy conditional distribution $f_\theta(\cdot|\bar{a})$ which is represented by a generative model, $g_\theta(z, \bar{a})$.

**(Semi)parametric density estimation.** Our framework uses a conditional generator, $g_\theta(z, \bar{a})$, to approximate the proxy conditional distribution $f_\theta(\cdot|\bar{a})$ under all $\bar{a}$. This differs from the parametric/semi-parametric density estimation approaches in Kennedy et al. (2023); Melnychuk et al. (2023), which directly estimates $f_\theta(\cdot|\bar{a})$ for each $\bar{a}$. The major advantage of our generative framework is its applicability in high-dimensional outcomes: it is computationally prohibitive to estimate high-dimensional densities. Therefore, we generate samples that represent this high-dimensional counterfactual distribution, instead of estimating it. This is a common practice in generative models, such as GAN (Goodfellow et al., 2014) and VAE (Kingma & Welling, 2013), where the generator learns to generate diverse image samples without estimating the density underlying the image distributions. Another advantage of our approach is that it trains only a single model for all treatment combinations, and has the potential to generalize to continuous treatments. Furthermore, our Proposition 1 is more sample efficient as compared to the parametric/semi-parametric density estimation approaches. The framework in Melnychuk et al. (2023), when extended to the time-varying scenario using IPTW, requires integrating the log-likelihood of the density model over both the observed samples and the outcome space $\mathcal{Y}$, see (10). In practice, this will require performing a Monte Carlo sampling of $Y$ for each gradient step to optimize (10), which can be prohibitive when $\mathcal{Y}$ is high-dimensional. Our proposed loss function in Proposition 1, on the other hand, only requires computing the weighted log-likelihood over observed samples which is easy to implement. Therefore, our Proposition 1 can be viewed as a novel reformulation of (10) into a generative training framework that enhances the scalability of model training for high-dimensional outcomes.

$$\mathbb{E}_{y \sim f_{\bar{a}}}[-\log f_\theta(y)] \approx \int_{y \in \mathcal{Y}} \log f_\theta(y) \frac{1}{N} \sum_{(y,\bar{a},\bar{x}) \in \mathcal{D}} w_\phi(\bar{a}, \bar{x}) f(y, \bar{a}, \bar{x}) dY. \tag{10}$$

**Doubly-robust density estimators.** Doubly-robust density estimators have proven successful in directly estimating the counterfactual density in the static setting (Kennedy et al., 2023; Melnychuk et al., 2023). In theory, one may extend our IPTW-based framework to a doubly-robust framework to ensure robustness. We opted for an IPTW only based framework because of practical challenges in combining IPTW with a outcome-based model, such as G-computation (Robins & Hernan, 2008; Li et al., 2021), for high-dimensional outcomes. When $Y$ is potentially high-dimensional, correct estimation of the outcome model using methods such as G-net might be challenging. This is due to the need to estimate both the conditional outcome distribution $f(Y_t|\bar{X}_t, \bar{A}_t)$ and the conditional covariate distribution $f(X_t|\bar{X}_{t-1}, \bar{A}_{t-1})$, where the first term involves a high-d outcome and the second term involves estimating a continuous density. As a result, we have also shown that the outcome-based method, `G-Net`, underperforms than our IPTW based framework. Therefore, we opted for the IPTW-based approach in proposition 1. We remain open about the possibility of adopting the doubly-robust approach for future work, especially if insights emerge regarding the development of an outcome-based method for accurately approximating high-dimensional, time-varying counterfactual distributions.

## F DERIVATION AND IMPLEMENTATION DETAILS OF VARIATIONAL LEARNING

**Derivation of the proxy conditional distribution** Now we present the derivation of the log conditional probability density function (PDF) in (7). To begin with, it can be written as:

$$\log f_\theta(y|\bar{a}) = \log \int p_\theta(y, z|\bar{a}) dz,$$

where $z$ is a latent random variable. This integral has no closed form and can be usually estimated by Monte Carlo integration with importance sampling, *i.e.*,

$$\int p_\theta(y, z|\bar{a}) dz = \mathbb{E}_{z \sim q(\cdot|y, \bar{a})} \left[ \frac{p_\theta(y, z|\bar{a})}{q(z|y, \bar{a})} \right].$$

Here $q(z|y,\overline{a})$ is the proposed variational distribution, where we can draw sample $z$ from this distribution given $y$ and $\overline{a}$. Therefore, by Jensen's inequality, we can find the evidence lower bound (ELBO) of the conditional PDF:

$$\log f_\theta(y|\overline{a}) = \log \mathop{\mathbb{E}}_{z\sim q(\cdot|y,\overline{a})} \left[ \frac{p_\theta(y,z|\overline{a})}{q(z|y,\overline{a})} \right] \geq \mathop{\mathbb{E}}_{z\sim q(\cdot|y,\overline{a})} \left[ \log \frac{p_\theta(y,z|\overline{a})}{q(z|y,\overline{a})} \right].$$

Using Bayes rule, the ELBO can be equivalently expressed as:

$$
\begin{aligned}
\mathop{\mathbb{E}}_{z\sim q(\cdot|y,\overline{a})} \left[ \log \frac{p_\theta(y,z|\overline{a})}{q(z|y,\overline{a})} \right] &= \mathop{\mathbb{E}}_{z\sim q(\cdot|y,\overline{a})} \left[ \log \frac{p_\theta(y|z,\overline{a})p_\theta(z|\overline{a})}{q(z|y,\overline{a})} \right] \\
&= \mathop{\mathbb{E}}_{z\sim q(\cdot|y,\overline{a})} \left[ \log \frac{p_\theta(z|\overline{a})}{q(z|y,\overline{a})} \right] + \mathop{\mathbb{E}}_{z\sim q(\cdot|y,\overline{a})} \left[ \log p_\theta(y|z,\overline{a}) \right] \\
&= -D_{\mathrm{KL}}(q(z|y,\overline{a})||p_\theta(z|\overline{a})) + \mathop{\mathbb{E}}_{z\sim q(\cdot|y,\overline{a})} \left[ \log p_\theta(y|z,\overline{a}) \right].
\end{aligned}
$$

**Implementation details**  For the KL-divergence term in the ELBO (7), both $q(z|y,\overline{a})$ and $p_\theta(z|\overline{a})$ are often modeled as Gaussian distributions, which allows us to compute the KL divergence of Gaussians with a closed-form expression. In practice, we introduce two additional generators, including the encoder net $g_{\mathrm{encode}}(\epsilon, y, \overline{a})$ and the prior net $g_{\mathrm{prior}}(\epsilon, \overline{a})$, respectively, to represent $q(z|y,\overline{a})$ and $p_\theta(z|\overline{a})$ as transformations of another random variable $\epsilon \sim \mathcal{N}(0, I)$ using reparameterization trick (Sohl-Dickstein et al., 2015). A common choice is a simple factorized Gaussian encoder. For example, the approximate posterior $q(z|y,\overline{a})$ can be represented as:

$$q(z|y,\overline{a}) = \mathcal{N}(z; \mu, \mathrm{diag}(\Sigma)),$$

or

$$q(z|y,\overline{a}) = \prod_{j=1}^{r} q(z_j|y,\overline{a}) = \prod_{j=1}^{r} \mathcal{N}(z_j; \mu_j, \sigma_j^2).$$

The Gaussian parameters $\mu = (\mu_j)_{j=1,\dots,r}$ and $\mathrm{diag}(\Sigma) = (\sigma_j^2)_{j=1,\dots,r}$ can be obtained using reparameterization trick via an encoder network $\phi$:

$$
\begin{aligned}
(\mu, \log \mathrm{diag}(\Sigma)) &= \phi(y, \overline{a}), \\
z &= \mu + \sigma \odot \epsilon,
\end{aligned}
$$

where $\epsilon \sim \mathcal{N}(0, I)$ is another random variable and $\odot$ is the element-wise product. Because both $q(z|y_i, \overline{a}_{i-1})$ and $p_\theta(z|\overline{a}_{i-1})$ are modeled as Gaussian distributions, the KL divergence can be computed using a closed-form expression.

The log-likelihood of the second term can be implemented as the reconstruction loss and calculated using generated samples. Maximizing the negative log-likelihood $p_\theta(y|z,\overline{a})$ is equivalent to minimizing the cross entropy between the distribution of an observed outcome $y$ and the reconstructed outcome $\widetilde{y}$ generated by the generative model $g$ given $z$ and the history $\overline{a}$.

We emphasize that our model is not tied to any specific type of generative models and learning algorithms, and we use the variational learning framework for illustrative purposes.

## G  ADDITIONAL EXPERIMENT DETAILS

### G.1  BASELINES

Here we present an additional review of each baseline method in the paper as well as implementation details.

**Marginal structural model with a fully-connected neural network (`MSM+NN`)**  We include the classic `MSM+NN` proposed in (Robins et al., 1994; Robins, 1986). This classical framework assumes that the counterfactual mean of the outcome variable can be represented as a linear function of the treatments. We use this model while replacing the linear model with a 3-layer fully-connected neural

network, $g_{\mathrm{msm}}$. This serves as a deterministic baseline for our generative framework. We learn the `MSM+NN` using stochastic gradient descent with a weighted loss function:

$$\sum_{(y,\overline{a},\overline{x})\in\mathcal{D}} w_\phi(\overline{a},\overline{x})(y - g_{\mathrm{msm}}(\overline{a}))^2.$$

To establish a fair comparison, we train the `MSM+NN` using an identical training size to that of the `MSCVAE` model. We train the `MSM+NN` for $1,000$ epochs with a learning rate of $0.01$. However, it's important to note that in this particular setup, our capacity is limited to estimating the mean instead of the entire distribution. For computing the Wasserstein distance in the full-synthetic experiments, we treat the `MSCVAE` samples as coming from a degenerate distribution at its predicted value.

**Conditional variational autoencoder (`CVAE`)**   To examine the impact of Inverse Probability of Treatment Weighting (IPTW) on training generative models, we include a vanilla conditional variational autoencoder (`CVAE`) with an architecture identical to that of the `MSCVAE`, but excluding IPTW weighting. The `CVAE` is a widely-used type of conditional generative model that has found applications in various tasks, including image generation (Mishra et al., 2018; Sohn et al., 2015), neural machine translation (Pagnoni et al., 2018), and molecular design (Lim et al., 2018b). To train the `CVAE`, we follow the same procedure as `MSCVAE`, with the exception that we replace the loss function with the unweighted version of (5).

$$\sum_{(y,\overline{a},\overline{x})\in\mathcal{D}} \log f_\theta(y|\overline{a}),$$

where $f_\theta(\cdot)$ is the conditional distribution represented by the `CVAE`.

**Kernel density estimator (`KDE`)**   We use a Gaussian kernel density estimator (Rosenblatt, 1956) to estimate the empirical conditional distribution from the observed data. This is achieved by running KDE on the observed outcomes with the same treatments, *i.e.*,

$$f_{\overline{a}} \approx g_{\mathrm{kde}}(y|\overline{A} = \overline{a}),$$

where $g_{\mathrm{kde}}(\cdot)$ is the KDE estimator. We learn the KDE with bandwidth set to $0.5$, $1$, $1.5$, and $2$, respectively, and report the metrics with bandwidth $= 0.5$ as the optimal results.

**Semi-parametric Plug-in method based on pseudo-population (`Plugin+KDE`)**   We include a baseline using Lemma 1 as a plugin estimator by following the semi-parametric KDE approach in Melnychuk et al. (2023). Specifically, we rewrite Lemma 1 as:

$$f_{\overline{a}}(y) \approx \sum_{(y,\overline{a},\overline{x})\in\mathcal{D}} \mathbb{1}\{\overline{A} = \overline{a}\} w_\phi(\overline{a},\overline{x}) f\left(y, \overline{A}, \overline{x}\right).$$

To estimate the right-hand side of the equation, we performed KDE on $y|\overline{A} = \overline{a}$ where each sample tuple $(y,\overline{a},\overline{x})$ is weighted by its IPTW, $w_\phi(\overline{a},\overline{x})$, for each $\overline{A} = \overline{a}$ separately. The bandwidth is set to be the same as in `KDE`.

**G-Net (`G-Net`)**   We implement `G-net` proposed in (Li et al., 2021) based on G-computation. For our experiment setting, at each time step $t \in [T]$, we designed the conditional covariates block, the history representation block, and the final conditional outcome block as a 3-layer fully connected neural network respectively. The types of blocks are interconnected to form sequential net structures across different time steps, followed by a conditional outcome block at the end, which has a 2-layer structure. This makes the `G-net` model include a total of $(2 \times d) + 1$ blocks. The loss function is the sum of the mean squared error:

$$\sum_{(\overline{x},y)\in\mathcal{D}} (\widehat{\overline{x}} - \overline{x})^2 + (\widehat{y} - y)^2,$$

where $\widehat{\overline{x}}$ and $\overline{x}$ are the predicted and groundtruth covariate history, while $\widehat{y}$ and $y$ are the predicted and groundtruth outcome. Following the original literature, we impose a Gaussian parametric assumption over the underlying counterfactual distribution, and introduce prediction variability by adding Gaussian noise whose variance is empirically estimated from the residuals between the predicted and groundtruth outcomes.

## G.2 Experiment metrics

To quantify the quality of the approximated counterfactual distributions, we used the following metrics:

**Mean**   This is the difference between the empirical mean of the evaluated samples.

**1-Wasserstein Distance**   We used the earth mover's distance, which is defined as:

$$l_1(u, v) = \inf_{\pi \in \Gamma(u,v)} \int_{\Omega \times \Omega} |x - y| d\pi(x, y),$$

where $\Gamma(u, v)$ is the joint probability distributions for the groundtruth and learned counterfactual distributions, and $\Omega$ is the space of each distribution.

**FID\***   Both semi-synthetic datasets have high-dimensional outcomes, making comparisons using the mean or Wasserstein distance of the distributions less interpretable. A common approach in the generative model community is FID (Fréchet inception distance ). In summary, FID uses a pre-trained neural network (frequently the inception v3 model) to obtain a feature vector for each sample, generated for groundtruth. The feature vector is the activation of the last pooling layer prior to the output layer of the pre-trained network. The feature vectors are then summarized as multivariate Gaussians by computing their mean and covariances. The distance between the generated or groundtruth image distribution is then computed by calculating the 2-Wasserstein distance between two sets of Gaussians. A lower FID score represents a more realistic distribution for the generated images.

Since FID is not specifically designed for our TV-MNIST and semi-synthetic COVID-19 datasets, we propose to use FID* by following a similar idea of FID. For the semi-synthetic COVID-19 dataset, we first compute a PCA projection matrix of size $67 \times 2$ using samples from the counterfactual distribution under each treatment. The projection serves as the purpose of the pre-trained network in the original FID because it captures key information, including spatial correlation, of the 67-dimensional outcome variables. For each treatment combination, we then project the 67-dimensional samples into the 2-dimensional representational space using the PCA projection matrix and compute the 1-Wasserstein distance of the projection between the generated and counterfactual samples. A lower FID* score represents the generated samples have a similar distribution compared to the counterfactual ones.

For the TV-MNIST dataset, we use a 3-layer fully-connected neural network pre-trained to classify MNIST images. This network serves as the purpose of the pre-trained network in the original FID because it represents the semantic information (the digit label) of the 784-dimensional outcome variables. For each treatment combination, we then project the 784-dimensional samples into a 1-dimensional label space using the pre-trained MNIST classifier and compute the 1-Wasserstein distance of the projection between the generated and counterfactual samples. A lower FID* score represents the generated samples have a similar semantic distribution (in terms of the digit labels) compared to the counterfactual ones.

## G.3 Experiment set-up

The counterfactual generator $g_\theta$, the IPTW $w_\phi$, and the encoder network $g_{\text{encode}}$ share the same two-layer fully-connected network architecture with ReLU activation. The layer width is set to $1,000$, and the length of the latent variable $z$ is set to $r$ which is determined by the specific synthetic experiment setting: $r = 5$ for $d = 1$ and $d = 3$, $r = 10$ for $d = 5$ and all the semi-synthetic and real data. For $g_{\text{encode}}$, the fully-connected networks map the $d + 1$ dimensional input vector (consisting of a $d$-dimensional treatment and 1-dimensional response) to the $r$-dimensional latent representation. For $g_\theta$, the fully-connected networks map the $r + d$ dimensional input vector (consisting of a $d$-dimensional treatment and $r$-random noise) to the 1-dimensional generated counterfactual outcome. For $w_\phi$, the fully-connected networks map the $2d$-dimensional input vector (consisting of a $d$-dimensional treatment and $d$-dimensional covariate) to the 1-dimensional conditional probability. We use a Sigmoid output layer for $w_\phi$ to ensure the output falls within $[0, 1]$. We set the batch size to 256 and the number of training epochs to 200 for training all the models in both synthetic and real data

Table 2: Coefficients of the linear model in synthetic data generation

| | $\alpha$ | $\beta$ | $\gamma$ |
|---|---|---|---|
| $d=1$ | $(-3, 2, -1)$ | $(-0.5, 0.5)$ | $(0)$ |
| $d=3$ | $(-1, 12, 6, 3, 2, 1, 0.5)$ | $(-0.5, 0.5, -0.5, 0.5, -0.5, 0.5)$ | $(-1, 1.5, 1, -1.5, -1)$ |
| $d=5$ | $(-1, 12, 6, 3, 1, 0.5, 2, 1, 0.5, 0.1, 0.05)$ | $(-0.5, 0.5, -0.5, 0.5, -0.5, 0.5, -0.5, 0.5, -0.5, 0.5)$ | $(-1, 1.5, 1, 0.5, 0.1, -1.5, -1, -0.5, -0.1)$ |

settings. The learning rate was set to $10^{-3}$ with a linear step-wise learning rate scheduler (Pytorch learning rate scheduler function `StepLR`) to ensure stable convergence of the learning process.

### G.4   FULLY SYNTHETIC DATA

In this section, we provide an overview of the procedures for generating synthetic data. Our goal is to evaluate the performance of the proposed `MSCVAE` method and compare it to baseline approaches in the context of time-varying treatments. We follow the classic setting in (Robins et al., 1999) and simulate time series data with time-varying treatments and covariates. The presence of the time-varying confounders serves as an appropriate testbed for comparing MSM-based models to the baselines. To be specific, we generate three synthetic datasets with varying levels of historical dependence denoted as $d$. Each dataset consists of 10,000 trajectories, which represent recorded observations of individual subjects. These trajectories comprise 100 data tuples, encompassing treatment, covariate, and outcome values at specific time points. The causal relationships between these variables are visually depicted in Figure 8. For each time trajectory of length $T$, the datasets are generated based on the following equations:

$$X_0 \sim \text{uniform}(0, 1), \tag{11}$$

$$X_t = \gamma_0 + \sum_{\tau=t-d+1}^{t-1} \gamma_{t-\tau} A_\tau + \sum_{\tau=t-d+1}^{t-1} \gamma_{d+t-\tau-1} X_\tau, \tag{12}$$

$$\mathbb{P}\{A_t = 1\} = \sigma(\beta_0 + \sum_{\tau=t-d+1}^{t-1} \beta_{t-\tau} A_\tau + \sum_{\tau=t-d+1}^{t} \beta_{d+t-\tau-1} X_\tau), \tag{13}$$

$$Y_t = \alpha_0 + \sum_{\tau=t-d+1}^{t} \alpha_{t-\tau} A_t + \sum_{\tau=t-d+1}^{t} \alpha_{d+t-\tau} X_\tau + \epsilon, \tag{14}$$

where $\epsilon \sim \mathcal{N}(0, 0.05)$ is the observation noise and $\sigma(\cdot)$ is a Sigmoid function. The specific coefficients are set according to the values in Table 2 to ensure the generation of valid synthetic data distributions with diversity:

Adjusting $\beta_0$ will change the balance of the treatment combinations: when keeping the remaining $\beta$ coefficients, treatment variables $\bar{a}$, and covariates $\bar{x}$ unchanged, a smaller value of $\beta_0$ reduces the probability of treatment exposure, *i.e.*, $\mathbb{P}(A_t = 1)$. Consequently, this lower probability of treatment exposure results in a decrease in the occurrence of treatment combinations with exposures, leading to an imbalanced ratio among different treatment combinations. In Figure. 4, we set $\beta_0 = -0.5$ which results in an approximated balanced number of samples per treatment combination. In Appendix H, we include a figure by setting $\beta_0 = -2$, as a visualization of imbalanced treatment combinations.

To ensure the validity of our synthetic data generation process, we verify that the three assumptions outlined in Appendix A are satisfied. Assumptions 1 and 3 are naturally met because the ground truth model guarantees that the counterfactual outcome equals the observed outcome and that there are no unmeasured confounders. As for assumption 2, since the conditional probability of treatment is the Sigmoid function applied to a finite linear combination of historical treatments and covariates, it will always be positive.

Once the synthetic data is generated, we obtain counterfactual distributions to assess the performance of our proposed method. Specifically, we use the synthetic data to obtain samples from the counterfactual outcome distribution, $Y(\bar{a})$, for any given treatment combination $\bar{a}$. This is achieved by iteratively fixing the treatment sequence in the time series and generating the covariates and response variables according to equations (12) and (14) for each of the 10,000 trajectories. The detailed procedure for obtaining a single counterfactual outcome sample is summarized in Algorithm 2.

---

**Algorithm 2** Algorithm for obtaining a counterfactual sample
---

**Input:** Generated trajectory of a single subject: $\{(Y_t, X_t, A_t)\}_{t=1,\cdots,T}$.

**Initialization:** Given the treatment history $\overline{A}_T = \overline{a}$.

**for** $\tau = T - d + 1 : T$ **do**

    1. Generate the covariate $x_\tau$ based on $\overline{A}_{\tau-1}$ and $\overline{X}_{\tau-1}$ according to (12).

    2. Update the covariate $\overline{X}_\tau \leftarrow x_\tau$.

**end for**

Generate $Y(\overline{a})$ based on $\overline{A}_T$ and $\overline{X}_T$ according to (14).

**return** $Y(\overline{a})$

---

Table 3: Real data description

| Name | Description | Min | Max | Mean | Median | Std |
|------|-------------|-----|-----|------|--------|-----|
| $Y$ | county-wise incremental new cases count ($\log_{10}$) | 0 | $1.15 \times 10^{-1}$ | $2 \times 10^{-3}$ | $1 \times 10^{-3}$ | $2.7 \times 10^{-3}$ |
| $A$ | county-wise mask mandate | 0 | $1 \times 10^{0}$ | $5.35 \times 10^{-1}$ | $1 \times 10^{0}$ | $4.99 \times 10^{-1}$ |
| $X^{(0)}$ | county-wise incremental death cases count ($\log_{10}$) | 0 | $3.12 \times 10^{-3}$ | $3 \times 10^{-4}$ | 0 | $9 \times 10^{-5}$ |
| $X^{(1)}$ | county-wise average retail and recreation | $-5.45 \times 10^{1}$ | $2.23 \times 10^{1}$ | $-4.27 \times 10^{0}$ | $-3.33 \times 10^{0}$ | $6.16 \times 10^{0}$ |
| $X^{(2)}$ | county-wise symptom value | 0 | $3.23 \times 10^{1}$ | $9.3 \times 10^{-1}$ | $8.1 \times 10^{-1}$ | $5.1 \times 10^{-1}$ |

## G.5 SEMI-SYNTHETIC TIME-VARYING MNIST DATA

We provide a benchmark based on the MNIST dataset. Specifically, the outcomes are MNIST images ($m = 784$). First, we compute a one-dimensional summary, the $\phi$ score (Jesson et al., 2021), using each MNIST image. The $\phi$ value of an image depends on its average light intensity and its digit label. We refer the readers to Jesson et al. (2021) for the details on computing $\phi$. Here we set the length of history dependence, $d$, to 3. We then define a linear model of 1-dimensional latent process to G.4 and simulate $1,000$ trajectories of the $(X, A, Y)$ tuples of 100 time points according to the following equations:

$$X_0 \sim \text{uniform}(0, 1), \tag{15}$$

$$X_t = \gamma_0 + \sum_{\tau=t-2}^{t-1} \gamma_{t-\tau} A_\tau + \sum_{\tau=t-2}^{t-1} \gamma_{t-\tau+2} X_\tau, \tag{16}$$

$$\mathbb{P}\{A_t = 1\} = \sigma(\beta_0 + \sum_{\tau=t-2}^{t-1} \beta_{t-\tau} A_\tau + \sum_{\tau=t-2}^{t} \beta_{t-\tau+2} X_\tau), \tag{17}$$

$$\phi_t = 0.5 \left\lceil 10\sigma(\alpha_0 + \sum_{\tau=t-2}^{t} \alpha_{t-\tau} A_t + \sum_{\tau=t-2}^{t} \alpha_{t-\tau+3} X_\tau) - 0.6 \right\rceil, \tag{18}$$

$$Y_t \sim \{\text{MNIST}(i) : i = \arg\min |\phi_i - \phi_t|\}, \tag{19}$$

where $\sigma(\cdot)$ is a Sigmoid function, $\lceil \cdot \rceil$ is the ceiling function, and $\text{MNIST}(i)$ represents the MNIST image indexed by $i$. The coefficients are set according to Table 2 to ensure the generation of diverse data distributions. We generate the counterfactual samples according to Algorithm 2 by replacing the corresponding propensity and outcome models with the formulations above. The generated observations and counterfactual samples under the same treatment combinations may correspond to MNIST images of different labels. This way we can qualitatively assess the performance of an algorithm by comparing the labels of the MNIST images it generates against the counterfactual samples, as in as in Figure. 5.

## G.6 COVID-19 DATA

Since both the semi-synthetic Pennsylvania COVID-19 mask data and the real nationwide COVID-19 mask datasets are based on the same set of aggregated sources. We first introduce the data sources and then include the details of each dataset respectively.

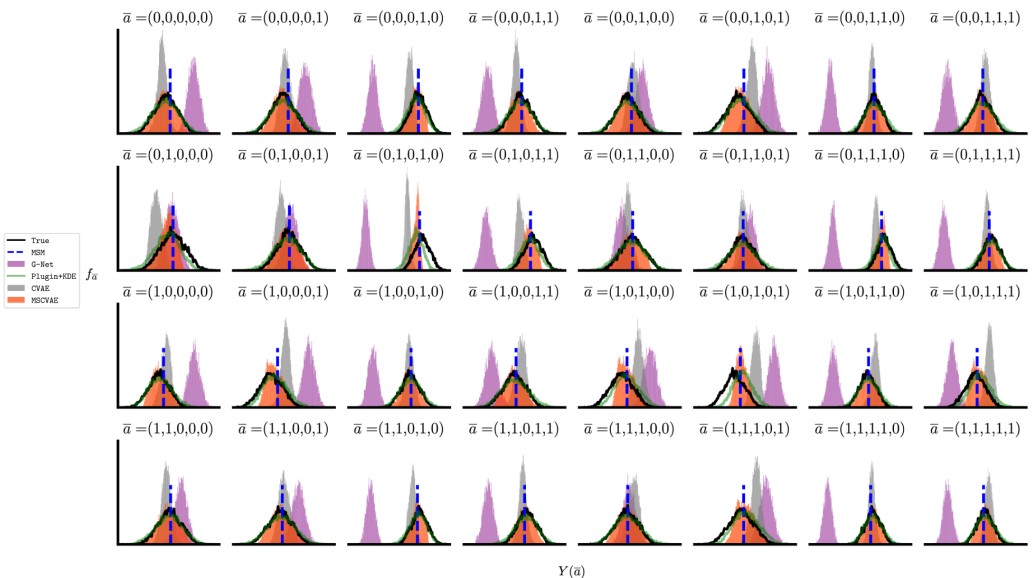

Figure 9: The estimated and true counterfactual distributions for $d = 5$ on synthetic datasets.

The real data used in this study comprises COVID-19-related demographic statistics collected from $3,219$ counties across 56 states/affiliated regions of the United States. The data covers a time period from 2020 to 2022. We obtained the data from reputable sources including the U.S. Census Bureau (Bureau, 2022), the Center for Disease Control and Prevention (for Disease Control, 2021), Google (Google, 2022), the CMU DELPHI group's Facebook survey (Group, 2022), and the New York Times (Times, 2021). To capture a relevant time window for analysis, we set the history dependence length $d$ to 3, as most COVID-19 symptoms tend to subside within this timeframe (Maltezou et al., 2021).

In our analysis, the treatment variable $A$ is the state-wise mask mandate indicator variable. A value of 0 indicates no mask mandate, while a value of 1 indicates the enforcement of a mask mandate. Notably, we observe a pattern in the data where mask mandates are typically implemented simultaneously across all counties within a state. This synchronization justifies the use of the state-wise mask mandate count as the treatment variable. As for the covariates $X$, we choose the county-wise incremental death count, state-wise the average retail and recreation metric (representing changes in mobility levels compared to a baseline, which can be negative), the state-wise symptom value, and the state-wise vaccine dosage.

**Pennsylvania COVID-19 mask mandate data** For the semi-synthetic dataset, we specifically look at the data within the state of Pennsylvania because of its long records spanning 106 weeks from 2020 to 2021. We set the four state-level covariates (per 100K people): the number of deaths, the average retail and recreation mobility, the surveyed COVID-19 symptoms, and the number of administered COVID-19 vaccine doses. We set the county-level incremental death count to the state level by computing a state average. We set the state-level mask mandate policy as the treatment variable, and the county-level number of new COVID-19 cases (per 100K) as the outcome variable, resulting in $m = 67$ since there are 67 counties in the state of Pennsylvania. We simulate $2,000$ trajectories of the $(X, A, Y)$ tuples of 300 time points (each point corresponding to a week) according to the following formula:

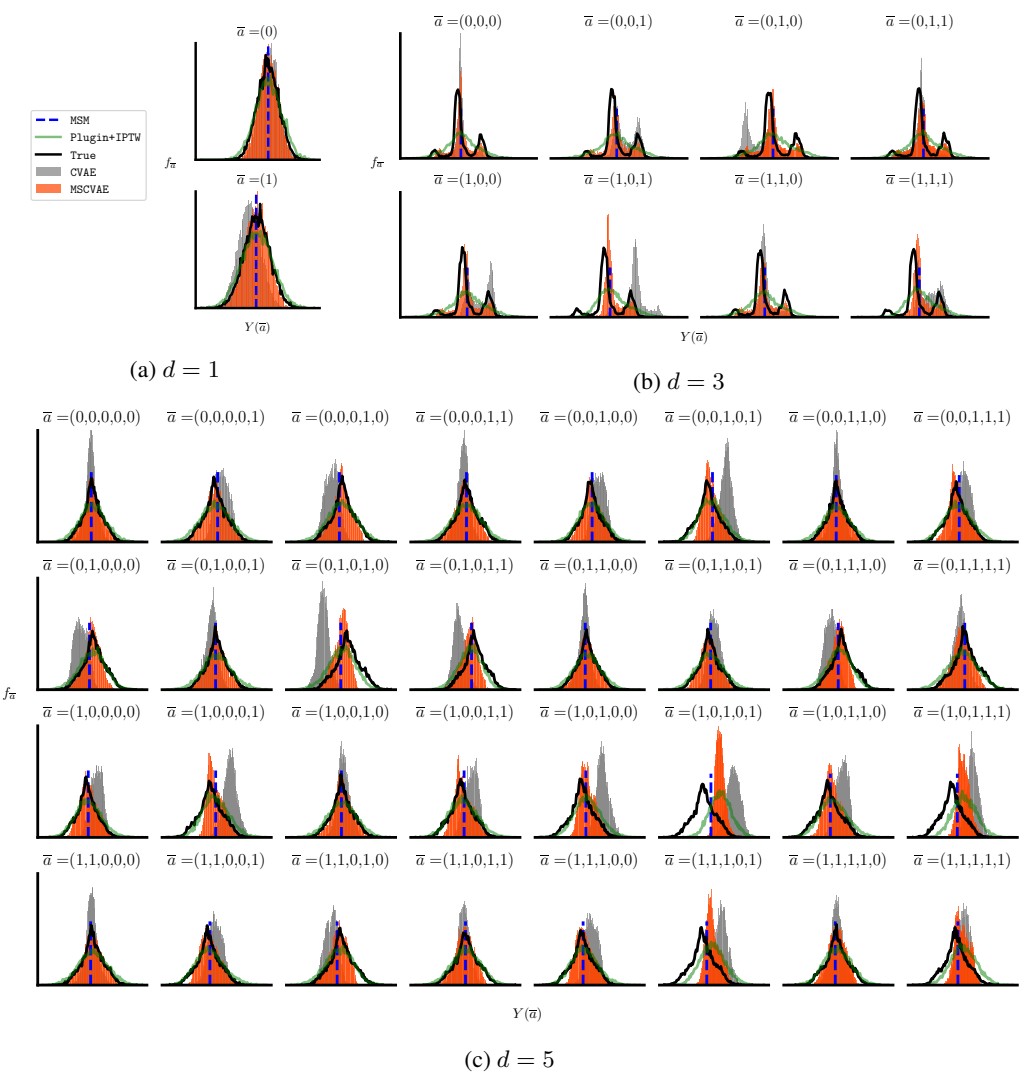

Figure 10: The estimated and true counterfactual distributions across various lengths of history dependence ($d = 1, 3, 5$) on synthetic datasets with imbalanced proportions of different treatment ($\beta_0 = -2$). Each sub-panel provides a comparison for a specific treatment combination $\bar{a}$. We exclude KDE and G-Net for illustrative purposes.

$$X_0 \sim \text{Real-World}(\cdot), \tag{20}$$

$$X_t = \hat{\mathbb{P}}(X_t | \overline{A}_t, \overline{X}_t), \tag{21}$$

$$\mathbb{P}\{A_t = 1\} = \sigma(\beta_0 + \sum_{\tau=t-2}^{t-1} \beta_{t-\tau} A_\tau + \sum_{\tau=t-2}^{t} \beta_{t-\tau+2} X_\tau), \tag{22}$$

$$Y_t^{base} = -0.2 A_{t-2} - 0.15 A_{t-1} - 0.1 A_t + 0.45 + \epsilon, \tag{23}$$

$$\mathbb{P}(L_t = 1) = \text{Bernoulli}(\prod_{j=1}^{4} X_\tau(j)), \tag{24}$$

$$Y_t(s) = Y_t^{base} + \begin{cases} \log(\mathcal{N}(s, \mu = [40.009, -75.133]^T, \Sigma = \mathbf{I})); & \text{if } L_t = 1, \\ \log(\mathcal{N}(s, \mu = [40.470, -79.980]^T, \Sigma = \mathbf{I})); & \text{otherwise.} \end{cases}, \tag{25}$$

where $\hat{\mathbb{P}}(\cdot)$ is learned with a 2-layer fully-connected neural network using the real data, $\epsilon \sim \mathcal{N}(0, 0.001)$ is the observation noise, $s$ is the 2-dimensional coordinate of a entry (county) in $Y_t$, $\sigma(\cdot)$ is a Sigmoid function. All other coefficients are set according to Table 2 to ensure the generation of diverse data distributions. We generate the counterfactual samples according to Algorithm 2 by replacing the corresponding outcome models with the formulations above. In summary, the hotspot (mode of the $Y_t$ vector) is either Philadelphia ($L_t = 1$) or Pittsburgh ($L_t = 0$), where the probability depends on the covariates $\overline{X}_t$. The values in the entries of $Y_t$ follow the log-likelihood of a 2-dimensional isotropic Gaussian centered at the hotspot. As a result, the counterfactual and observed distributions will be bimodally distributed with different hotspot probabilities. We can then visually assess the performance of the models by comparing the distribution of the hotspot from the generated outcome samples to those of the counterfactual samples, as in Figure. 6.

**Nationwide COVID-19 Mask data**  We perform a case study using real data by looking at the aggregated COVID-19 data sources from 2020 to 2021 spanning 49 weeks due to the limited availability of the nationwide data. We exclude 89 counties with zero incremental new cases count. These counties either do not have a significant amount of infectious cases or have small populations, leading to 3,130 counties across 56 states/affiliated regions of the United States. For variables that only have state-level records, we map them to the county level for simplicity.

We analyze the same set of variables as the semi-synthetic COVID-19 dataset but exclude the vaccine dosage covariate because of missing data in some states. To align the outcome variable with the covariates and treatment, we set it to measure one week after these variables. Due to the long-tailed distribution of the outcome variable, we apply a base-10 logarithmic transformation during the modeling process. Further details regarding the variables can be found in Table 3. We use the same model architecture described in Appendix G.3 to train the IPTW network and the MSCVAE. We generate counterfactual outcomes for treatment combinations $\overline{a} = (0, 0, 0)$ and $\overline{a} = (1, 1, 1)$. Since other treatment combinations occur rarely (less than 5% of observations), we exclude them from the final results.

## H  ADDITIONAL SYNTHETIC RESULTS

In the main paper, we presented a visual comparison of the learned counterfactual distributions and the true counterfactual distribution for various scenarios ($d = 1, 3$), as shown in Figure 4. Here, in Figure 9 we show the case for $d = 5$. We also provide a similar comparison while setting $\beta_0 = -2$ (as opposed to $\beta_0 = -0.5$,) where the treatment combinations are imbalancedly distributed (Figure 10). Consistent with the findings in Figure 4, our results in Figures 9 and 10 demonstrate the superior performance of the MSCVAE model (represented by the orange shading) in accurately capturing the shape of the true counterfactual distributions (represented by the black line) across all scenarios. This observation further validates the quantitative comparisons presented in Table 1, where MSCVAE achieves the smallest mean and Wasserstein distance among all baseline methods. These results highlight that our algorithm attains competitive performance even when certain treatment combinations occur less frequently compared to others. This situation is common in real-life scenarios where certain treatment combinations are favored due to factors such as policy inertia.

## I  GENERATING SAMPLES FROM CONDITIONAL COUNTERFACTUAL DISTRIBUTION

A related extension of our work might be to infer conditional counterfactual outcomes (related to the conditional average treatment effect, CATE). This corresponds to looking at the distribution of the outcome under a specific subpopulation. Since our covariates are assumed to be time-varying, a common approach is to introduce a set of static baseline covariates, $V$ (Robins et al., 1999). The baseline covariate $V$ denotes the static feature (such as a patient's gender or age) that will influence both the time-varying covariates $X$ and the outcome $Y$. We can then draw counterfactual samples from a specific sub-population by conditioning on the values of the $V$. In this framework, we observe $(Y_t^i, \overline{A}_t^i, \overline{X}_t^i, V^i)$, where $V \in \mathbb{R}^\nu$ is a static baseline variable tha differs by individual. Accordingly, the generator will have an additional input $g_\theta(z, \overline{a}, v) : \mathbb{R}^r \times \mathcal{A}^d \times \mathbb{R}^\nu \to \mathcal{Y}$, and the IPTW weights will

Table 4: Quantitative performance on fully-synthetic dataset with static baseline covariate

| | $V \in [-1,0]$ | | | | | | $V \in [0,1]$ | | | | | |
| | $d=1$ | | $d=3$ | | $d=5$ | | $d=1$ | | $d=3$ | | $d=5$ | |
| Methods | Mean ↓ | Wasserstein ↓ | Mean ↓ | Wasserstein ↓ | Mean ↓ | Wasserstein ↓ | Mean ↓ | Wasserstein ↓ | Mean ↓ | Wasserstein ↓ | Mean ↓ | Wasserstein ↓ |
|---|---|---|---|---|---|---|---|---|---|---|---|---|
| MSM+NN | **0.002 (0.003)** | 0.408 (0.408) | **0.064 (0.104)** | 0.449 (0.466) | **0.164 (0.441)** | 0.368 (**0.517**) | 0.015 (0.019) | 0.407 (0.412) | 0.057 (**0.068**) | 0.466 (0.475) | 0.182 (**0.484**) | 0.388 (**0.549**) |
| KDE | 0.194 (0.265) | 0.201 (0.266) | 0.557 (1.261) | 0.562 (1.261) | 0.562 (1.590) | 0.564 (1.590) | 0.216 (0.251) | 0.222 (0.256) | 0.547 (1.138) | 0.548 (1.138) | 0.559 (1.517) | 0.561 (1.517) |
| Plugin+KDE | 0.010 (0.013) | **0.117 (0.121)** | 0.073 (0.230) | **0.134 (0.230)** | 0.170 (0.823) | **0.196 (0.823)** | **0.009 (0.017)** | **0.121 (0.130)** | **0.055** (0.165) | **0.109 (0.165)** | **0.148** (0.670) | **0.182** (0.670) |
| G-Net | 0.431 (0.512) | 0.431 (0.512) | 0.819 (1.885) | 0.823 (1.885) | 0.815 (2.238) | 0.843 (2.238) | 0.452 (0.551) | 0.452 (0.551) | 0.738 (1.707) | 0.739 (1.707) | 0.735 (2.076) | 0.768 (2.076) |
| CVAE | 0.238 (0.327) | 0.240 (0.328) | 0.537 (1.192) | 0.571 (1.192) | 0.534 (1.478) | 0.585 (1.478) | 0.256 (0.290) | 0.258 (0.290) | 0.534 (1.094) | 0.565 (1.094) | 0.534 (1.423) | 0.591 (1.423) |
| MSCVAE | **0.009 (0.010)** | **0.051 (0.056)** | **0.058 (0.181)** | **0.083 (0.188)** | **0.169 (0.767)** | **0.186 (0.767)** | **0.006 (0.010)** | **0.047 (0.055)** | **0.046 (0.140)** | **0.068 (0.158)** | **0.151 (0.659)** | **0.172 (0.659)** |

\* Numbers represent the average metric across all treatment combinations and those in the parentheses represent the worst across treatment combinations.

become $w_\phi(\overline{a}, \overline{x}, v) = \frac{1}{\prod_{\tau=t-d+1}^{t} f_\phi(a_\tau | \overline{a}_{\tau-1}, \overline{x}_\tau, v)}$. Correspondingly, the objective in Proposition 1 will become $\mathbb{E}_v \; \mathbb{E}_{\overline{a}} \; \left[ \mathbb{E}_{y \sim f_{\overline{a}, v}} \; \log f_\theta(y | \overline{a}, v) \right] \approx \frac{1}{N} \sum_{(y, \overline{a}, \overline{x}, v) \in \mathcal{D}} w_\phi(\overline{a}, \overline{x}, v) \log f_\theta(y | \overline{a}, v)$.

We have conducted additional experiments for the fully synthetic data, where the baseline variable, $V$, was divided into two groups. The $V$ was uniformly drawn from $[-1, 0]$ in the first group and from $[0, 1]$ in the second group. We then looked at the performance of the generative samples corresponding to two sub-groups generating samples with the corresponding treatment combinations and the baseline covariate, $V$. Specifically, for the MSCVAE and CVAE, under a specific $\overline{a}$, we generated $v$ from $[-1, 0]$ and $[0, 1]$ uniformly at random and fed them into the conditional generators, along with the Gaussian noises, $z$. We have included the results in Table 4, Fig. 11 ($V \in [-1, 0]$), and Fig. 12 ($V \in [0, 1]$). It can be seen that *MSCVAE* still outperforms other baselines.

## J  CODE AVAILABILITY

We include a code demonstration for the MSCVAE in the supplementary materials in the code folder.

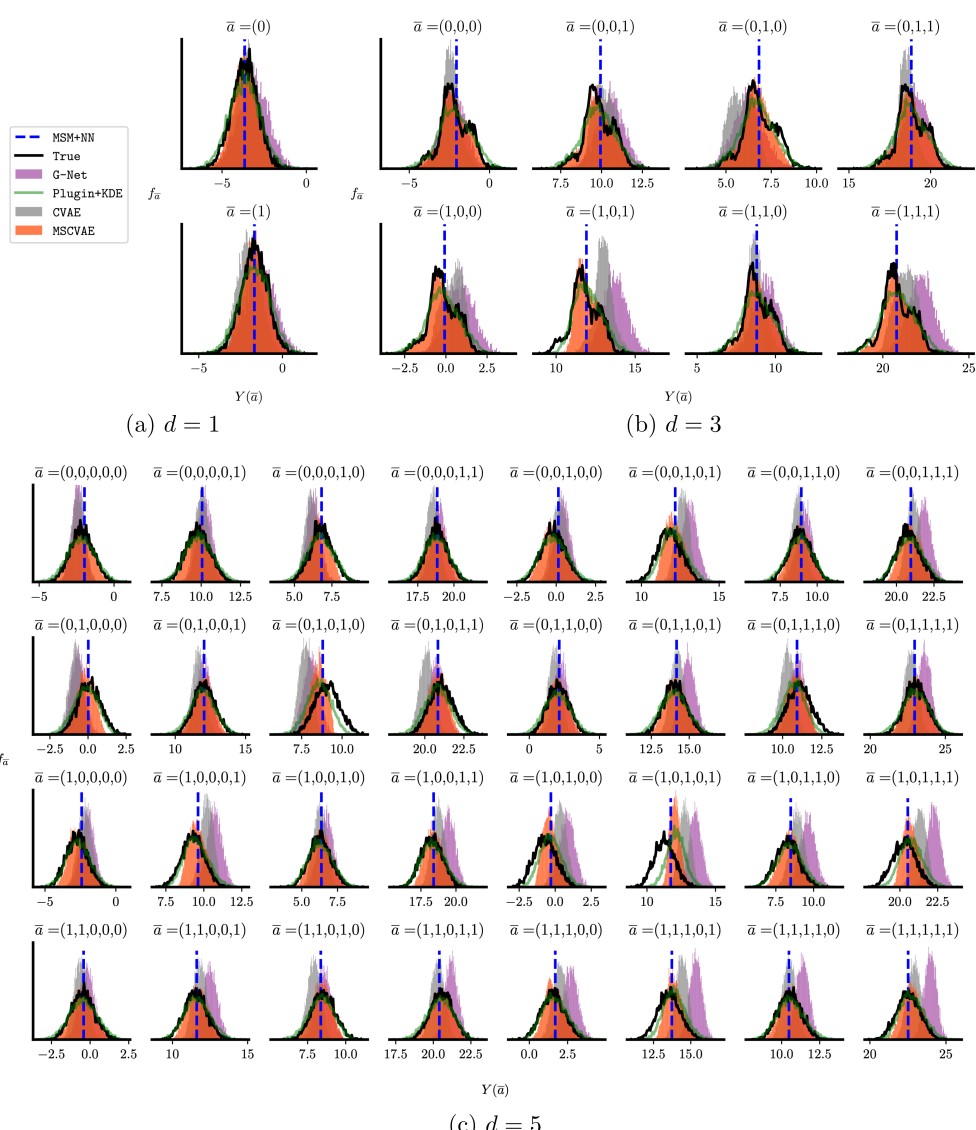

Figure 11: The estimated and true counterfactual distributions across various lengths of history dependence ($d = 1, 3, 5$) on synthetic datasets with $V \in [-1, 0]$. Each sub-panel provides a comparison for a specific treatment combination $\bar{a}$. We exclude `KDE` for illustrative purposes.

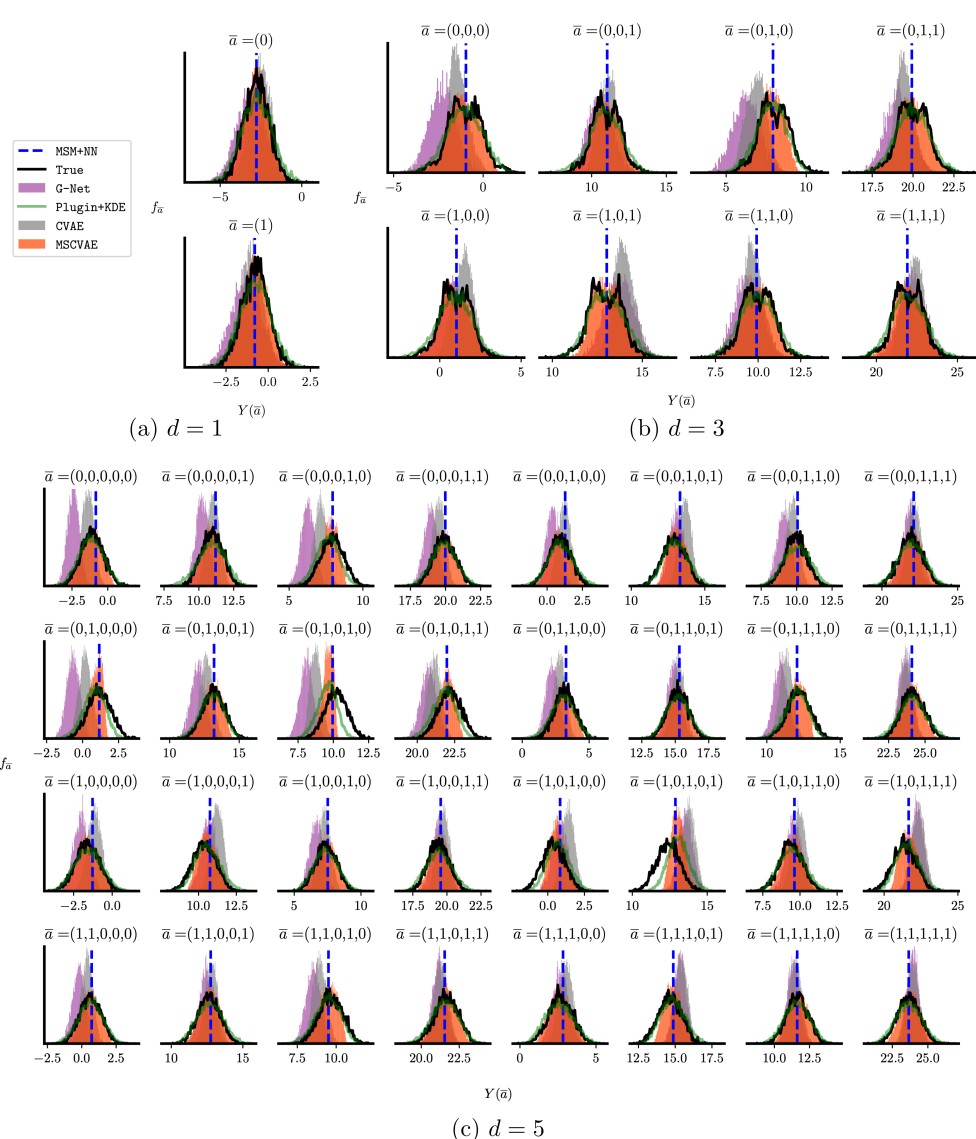

Figure 12: The estimated and true counterfactual distributions across various lengths of history dependence ($d = 1, 3, 5$) on synthetic datasets with $V \in [0, 1]$. Each sub-panel provides a comparison for a specific treatment combination $\overline{a}$. We exclude KDE for illustrative purposes.

