# OpenReview forum: "Counterfactual Generative Models for Time-Varying Treatments"
_ICLR.cc/2024/Conference — Submitted to ICLR 2024_

### Official Review · Reviewer_kkvR · 2023-10-27

**Soundness:** 3 good
**Presentation:** 2 fair
**Contribution:** 3 good
**Rating:** 6
**Confidence:** 3

**Summary:**

This study addresses an important issue of causal inference (counterfactual outcome in time-varying situation) by generating a counterfactual distribution. They conducted various experiments as well as getting good results. I think this is a nice paper. However, there are many parts of this paper where the interpretation needs to be improved. As I have been busy lately, it is possible that there are some details that I have not checked sufficiently.

**Strengths:**

1. This study proposes a new approach that can be used for high-dimensional outcomes. Most existing studies consider low-dimensional outcomes.
2. Addressing the issue of counterfactual outcomes by generating counterfactual distributions is interesting.
3. They conducted experiments on various datasets. Importantly, they used real data.
4. The experiment results provided by the authors are good.

**Weaknesses:**

1. Many parts of the explanation need to be improved. For example, authors focus on describing what they did and used, but not why they did it. How do readers use this model to solve causal issues, such as ITE estimation? In addition, for the description of datasets, readers may wonder what is the treatment in these datasets (authors only said "treatment variable").
2. Lack of comparison of technological innovations from previous approaches. It might be helpful to understand the contribution of this paper by adding a paragraph discussing this.

**Questions:**

1. How should readers use your methods to estimate ITE?
2. What do treatments represent in the datasets used in experiments?

---

> ### Author Response · Authors · 2023-11-19
> **Rebuttal by Authors**
>
> Thanks much for your valuable and insightful comments. We express our gratitude to the reviewer for your positive comments regarding the importance of addressing high-dimension outcomes, the use of generative models, and the positive experimental results. We also thank the reviewer for emphasizing the potential extension of our framework, the comparison to current literature, and the meaning of the treatment variables. We have updated the manuscript according to your comments. Please see our point-by-point responses below.
>
>
>  > How do readers use this model to solve causal issues, such as ITE estimation?
>
> We appreciate the reviewer's reference to Individual Treatment Effects (ITE). We would like to clarify that our approach to estimating the counterfactual distribution across a population, rather than individual-focused estimates (ITE), is driven by the practicality of our research's applications, such as in public health policy. Implementing policies on an individual basis can be costly and sometimes unfeasible. By focusing on the population level (or a sub-population), we capture individual differences, which is essential for effective policy-making. In contrast, the variation within individual counterfactuals often stems from observational noise and might need substantial data for precise estimation. Consider, for instance, the COVID-19 data: our objective is to analyze the counterfactual outcome of a state-wide policy (mask mandate) and provide policymakers with a range of possible counterfactual outcomes that reflect variability in county-level confirmed cases. In such contexts, policymakers might be less concerned with the estimation noise at the individual level and more focused on the broader variability across different individuals.
>
>  In the meantime, a related extension of our work might be to infer conditional counterfactual outcomes (related to the conditional average treatment effect, CATE). This corresponds to looking at the distribution of the outcome under a specific subpopulation. Since our covariates are assumed to be time-varying, a common approach is to introduce a set of static baseline covariates, $V$ (Robins et al, 1999). The baseline covariate $V$ denotes the static feature (such as a patient's gender or age) that will influence both the time-varying covariates $X$ and the outcome $Y$. We can then draw counterfactual samples from a specific sub-population by conditioning on the values of the $V$. In this framework, we observe ($Y_t^i$, $\overline{A}_t^i$, $\overline{X}_t^i$, $V^i$), where $V \in \mathbb{R}^{
>     \nu}$ is a static baseline variable that varies by individual.  Accordingly, the generator will have an additional input:
> $ g\_{\theta}(z, \overline{a},v): \mathbb{R}^r \times \mathcal{A}^d \times \mathbb{R}^{\nu} \rightarrow \mathcal{Y}$ , and the IPTW weights will become $ w\_\phi(\overline{a},\overline{x},v) = \frac{1}{\prod\_{\tau=t-d+1}^{t} f\_\phi(a\_{\tau}|\overline{a}\_{\tau-1},\overline{x}\_{\tau},v)}$.   Correspondingly, the objective in Proposition 1 will become $\mathbb{E}\_{v} ~\mathbb{E}\_{\overline{a}} ~\left[\mathbb{E}\_{y \sim f\_{\overline{a},v}} ~\log f\_\theta(y|\overline{a},v)\right] \approx \frac{1}{N}\sum\_{(y,\overline{a},\overline{x},v) \in \mathcal{D}}w\_\phi(\overline{a},\overline{x},v) \log f\_\theta(y|\overline{a},v).$
>
>  We have conducted additional experiments using the fully synthetic dataset, where the baseline variable, $V$, was divided into two groups. The $V$ was uniformly drawn from $[-1,0]$ in the first group and from $[0,1]$ in the second group. We then looked at the performance of the generative samples corresponding to two sub-groups. We included the results in **Appendix I** (Table 4, Fig.11, and Fig.12) of the revised manuscript.  Our method (MSCVAE) still consistently outperforms other baseline methods in the simplified results below (Wasserstein distance, smaller the better):
>
> - $V\in[-1,0]$
>
> |         | d=1  | d=3  |  d=5  |
> | ----------- | ----------- | ----------- |----------- |
> | MSM+NN      |   0.408    |  0.449   |  0.368 |
> | KDE               |      0.201 |  0.562 |  0.564  |
> | Plugin+KDE   |        0.117    |      0.134      |   0.196      |
> | G-Net             |        0.431        |      0.823       |   0.843    |
> | CVAE             |       0.240          |     0.571       |   0.585   |
> | MSCVAE        |        0.051         |      0.083      |    0.186   |
>
> - $V\in[0,1]$
>
> |         | d=1  | d=3  |  d=5  |
> | ----------- | ----------- | ----------- |----------- |
> | MSM+NN      |   0.407    |  0.466  |  0.388 |
> | KDE               |      0.222 |  0.548 |  0.561  |
> | Plugin+KDE   |        0.121   |      0.109      |   0.182      |
> | G-Net             |        0.452       |      0.739       |   0.768    |
> | CVAE             |       0.258          |     0.565      |   0.591   |
> | MSCVAE        |        0.047         |      0.068      |    0.172   |

---

> > ### Author Response · Authors · 2023-11-19
> >
> > > Lack of comparison of technological innovations from previous approaches. It might be helpful to understand the contribution of this paper by adding a paragraph discussing this.
> >
> > We are grateful to the reviewer for emphasizing the significance of comparing our approach with previous methodologies. In our manuscript, we have compared our methods with several state-of-the-art algorithms such as the G-Net and KDE based methods. We have also acknowledged the connection of our work to the broader literature in the Related Work section (1.1). Additionally, we delve into the detailed methodological connections to related methods in Appendix E. Following the valuable feedback received from reviewers, we have further refined and expanded Appendix E to clarify and enhance the comparison with existing approaches.
> >
> > > What do treatments represent in the datasets used in experiments?
> >
> > We appreciate the reviewer's attention to the meaning of the treatment variable in our datasets. As stated in the manuscript, for the semi-synthetic Pennsylvania COVID-19 dataset, the treatment refers to the implementation of state-level mask mandate policies. Similarly, in the real nationwide COVID-19 dataset, it denotes county-level mask mandate policies. Regarding the fully-synthetic data and the TV-MNIST datasets, these treatment variables represent abstract quantities, primarily used for validating the performance of our methods under different data settings. In these cases, the treatment could be conceptualized as hypothetical decision variables. For instance, in the context of the TV-MNIST dataset, it might represent actions like closing or opening one’s eye in relation to handwritten digits.

---

> > > ### Author Response · Authors · 2023-11-23
> > > **Follow-up**
> > >
> > > Dear reviewer,
> > >
> > > We sincerely hope that our responses have addressed your concerns. Please let us know if you have any additional questions or comments. We would be more than happy to follow up with additional details. Thanks again for taking the time to review our work.

---

### Official Review · Reviewer_QN9X · 2023-10-31

**Soundness:** 2 fair
**Presentation:** 3 good
**Contribution:** 2 fair
**Rating:** 5
**Confidence:** 4

**Summary:**

In this paper, the authors propose a method to estimate the high-dimensional counterfactual distributions for time-varying treatments. The method uses a generative model to do the task. The generative model allows for generating credible samples of the counterfactual outcomes given a time-varying treatment such that policymakers can assess a policy’s efficacy by exploring a range of probable outcomes and deepening their understanding of its counterfactual result.

**Strengths:**

The paper is technically sound. Generally, it is not hard for readers to follow. The ideas are presented well, but still, the clarity of the paper can be further improved.

**Weaknesses:**

Although readers should be able to follow and understand the notions presented in the paper, the paper is not organized well. For instance, the authors defer the standard causal assumptions to the Appendix.  The authors may not give detailed explanations about the causal assumptions in the main paper, but at least mention the names of the causal assumptions in the paper. Please refer to questions for further weakness.

**Questions:**

1. In Algorithm 1, the authors suggest that we should draw a sample epsilon from $N(0,I)$, where $I$ is the total number of individuals. What is the point of setting epsilon as a realization with large variance? Usually, $I$ can be very large. Indeed, if there are a large number of individuals, say $I=10000$, a realization can be very large. Further, why do you model epsilon as normally distributed?

2. I have a question about the training process. The objective function is given in Eqn. (2) which is approximated by Eqn. (5). Nevertheless, the computation is given for each t only, where t lies in between 1, … , T according to the paper. We can obtain T approximations according to Eqn. (5). During training, the goal is to minimize one objective value, but we can calculate T approximations where each of the T approximations can be thought of as the objective value. What should be the objective value of training?

3. In the paper, the authors state that t=1, …, T in the section of PROBLEM SETUP. However, when the authors present Algorithm 1, t = d, …, T. It is strange that t=d, …, T in Algorithm 1. Is it a typo mistake? If not, from my realization about Algorithm 1, d should be determined. How to determine the value of d?

---

> ### Author Response · Authors · 2023-11-19
> **Rebuttal by Authors**
>
> Thanks much for your valuable and insightful comments. We express our gratitude to the reviewer for your positive comments regarding the technical soundness of our method. We also thank the reviewer for the careful read and the mention of the confusion from the notations. We have updated the manuscript according to your comments. Please see our point-by-point responses below.
>
> > The authors may not give detailed explanations about the causal assumptions in the main paper, but at least mention the names of the causal assumptions in the paper.
>
> Thanks for the suggestion. In the revised manuscript, we mentioned the three causal assumptions in the methodology section.
>
> >  In Algorithm 1, the authors suggest that we should draw a sample epsilon from $N(0,I), where $I$ is the total number of individuals. What is the point of setting epsilon as a realization with large variance?
>
> We would like to clarify that $I$ should stand for the identity matrix, and $N$ represents the total number of samples. We have fixed it in the revised paper.
>
> >  Why do you model epsilon as normally distributed?
>
> The Gaussian assumption for the VAE is a common assumption which enables a closed form expression for the ELBO  (Kingma and Welling, 2013). To elaborate, our generative framework essentially involves identifying a transformation that converts random noise into the desired target distribution, which in this case is the counterfactual distribution. It's a standard practice in the field of generative models to use the normal distribution as the basis for our source distribution (Kingma and Welling, 2013; Sohn et al, 2015; Higgins et al, 2016).
>
> >  What should be the objective value of training?
>
> We would like to clarify that, the objective function in Proposition 1 is evaluated over the entire $N$ data samples, where $N$ represents the total number of samples across individuals and time (see below). Similarly, in the algorithm box, where batch optimization can be used, the objective is then evaluated over the mini-batch.
>
>  In the revised manuscript, we have clarified how data were indexed. Consistent with much of the literature on longitudinal causal inference, such as (Lim et al, 2018; Bica et al, 2020; Li et al, 2021), our approach assumes that data is observed across individuals, with each individual having data for $T$ time points. Importantly, given its history $\overline{X}$ and $\overline{A}$, the outcome, $Y$, can be treated as conditionally independent samples. Therefore, for simplicity, we could 'chop' the time series of each individual, leading to a total of $N$ data tuples across individuals and time. For example, if there are $1000$ individuals each with $100$  time points and $d=3$, then $N = 1000\times (100-3+1)=98,000$ (we truncate the first $d-1$ time points because they do not have a full history). Therefore, we denote the data tuples simply as ($y^i,\overline{x}^i,\overline{a}^i$), where $i$ denotes the index of the sample and ranges from $1$ to $N$.
>
> > It is strange that t=d, …, T in Algorithm 1. Is it a typo mistake?
>
> Following the previous point, we ranged $t$ from $d$ instead of $1$ because the first $d-1$ time points do not have full-length ($=d$) history.
>
> > How to determine the value of d?
>
> As a standard in longitudinal causal inference  (Robins et al, 1994; Lim et al, 2018; Bica et al, 2020; Li et al, 2021; Bica et al 2021), $d$ is pre-determined as prior knowledge of the data generating process.  This approach is validated by our numerical findings, proving to be an effective method. For instance, in our COVID-19 studies, we set $d=3$ following insights from epidemiological research, which indicates that a mask mandate typically requires two to three weeks to become effective in a population.
>
> Intuitively, $d$ can be considered as a causal structure that determines how many arrows should point to the current variable of interest from history. Therefore, if one is interested in estimating $d$ instead of pre-determining it, a potential method is causal discovery which automatically infer the DAG (Figure 8) from data.

---

> > ### Author Response · Authors · 2023-11-23
> > **Follow-up**
> >
> > Dear reviewer,
> >
> > We sincerely hope that our responses have addressed your concerns. Please let us know if you have any additional questions or comments. We would be more than happy to follow up with additional details. Thanks again for taking the time to review our work.

---

### Official Review · Reviewer_9P43 · 2023-11-01

**Soundness:** 3 good
**Presentation:** 3 good
**Contribution:** 2 fair
**Rating:** 6
**Confidence:** 4

**Summary:**

The author delved into examining the counterfactual results of treatments in dynamic treatment scenarios. They introduced a novel generative framework designed to produce counterfactual outcomes without explicitly learning the counterfactual distribution. In their approach, they put forth a learning objective that relies on reweighted Evidence Lower Bound (ELBO) within a conditional Variational Autoencoder (VAE), utilizing inverse propensity weights.

**Strengths:**

- Paper tackles the complex issue of estimating counterfactual outcomes in the face of time-varying treatment effects.
-The proposed method adeptly handles high-dimensional outcomes.
- Capable of generating counterfactual samples without imposing rigid assumptions on the distribution of the counterfactual outcome.

**Weaknesses:**

IPTW values can be notably small and, as highlighted by the author, require precise definition. This circumstance can exacerbate in sequential treatment scenarios.

Given we're handling a treatment sequence, it's important to note that the counterfactual treatment is not unique. However, the notation used does not reflect this.

I believe the following two papers could also serve as baseline references:
1. "Disentangled Counterfactual Recurrent Networks for Treatment Effect Inference Over Time"
2. "Estimating Counterfactual Treatment Outcomes Over Time Through Adversarially Balanced Representations"

**Questions:**

If x isn't utilized in the generator, what's the rationale for calculating weights based on x? Why not solely estimate treatment probability based on the treatment sequence?

Can you provide a proof for equation 2? What would be the difference in objective if we were to define the distance in terms of Maximum Mean Discrepancy (MMD) or Wasserstein distance?

Considering your framework, it appears straightforward to expand it to the individual outcome level. What led to the decision to overlook that possibility?

---

> ### Author Response · Authors · 2023-11-19
> **Rebuttal by Authors**
>
> Thanks much for your valuable and insightful comments. Our sincere thanks to the reviewer for recognizing the underlying motivation and flexibility of our approach. Please see our point-by-point responses below.
>
> > IPTW values can be notably small and, as highlighted by the author, require precise definition.
>
> Thank you for your comments regarding our use IPTW. We agree that IPTW can be unstable and might lead to less reliable results in certain contexts.  Nevertheless, it's important to highlight that integrating IPTW with our novel generative framework represents a significant and novel methodological advancement. This combination is particularly effective in capturing complex, high-dimensional patterns in counterfactual outcomes, an area our study uniquely addresses to the best of our knowledge.
>
> To address the stability concerns in a practical context, our experiments incorporated established stabilization strategies, such as quantile truncation and standardization, as extensively discussed in the literature (Xiao et al, 2010; Chesnaye et al, 2022). The results from our experiments also show that incorporating these techniques into the design of our marginal structure model greatly results in promising and dependable numerical outcomes.
>
> > Given we're handling a treatment sequence, it's important to note that the counterfactual treatment is not unique.
>
> We would like to clarify that our generator approximates the counterfactual distribution for any $\overline{a}$, so there should be an expectation over $\overline{a}$ for the objective function:
>     $$\hat{\theta}
>     = argmin_{\theta \in \Theta}~\mathbb{E}_{\overline{a}} ~\left[D_f(f\_{\overline{a}}, f\_\theta(\cdot|\overline{a})) \right],$$
>
> where $D$ represents the distributional distance between $f\_{\overline{a}}$ (true counterfactual distribution) and $f\_\theta$ (the proxy conditional distribution represented by our proposed counterfactual generator $g\_\theta$), such as KL divergence. We genuinely appreciate the reviewer's careful read of these details, and have updated the manuscript (Equation 2, 3 and Proposition 1) to address this issue.
>
> >  I believe the following two papers could also serve as baseline references ...
>
>  We thank the reviewer for mentioning these baseline methodologies. However, it's important to note that these methods are tailored for mean prediction, which differs from our objective of distribution estimation. Additionally, these methods primarily focus on predicting outcomes at the individual level, whereas our research is geared towards assessing the outcomes at the population level. These fundamental differences might make it challenging for a fair comparison between their methods and ours.  In this regard, we could incorporate these methods to compare with our method in terms of mean estimate, should the reviewer consider it necessary.
>
> > If x isn't utilized in the generator, what's the rationale for calculating weights based on x? Why not solely estimate treatment probability based on the treatment sequence?
>
>  We would like to clarify that, estimating treatment probability based solely on the treatment sequence will introduce bias. This is because the propensity score, $f(A_t|\overline{A}\_{t-1},\overline{X}\_{t})$, depends the history of both treatments and covariates. Therefore, IPTW can vary for the same treatment sequence, $\overline{a}$, owing to the variability in $X$. Importantly, the covariates $X$ play a crucial role in the standard approaches for IPTW estimation (Robins et al, 1999; Lim et al, 2018). As a result,  while we do not have to directly use  $X$ during training our conditional generative model, we still access them through the IPTW weights in Proposition 1.

---

> ### Author Response · Authors · 2023-11-19
> **Rebuttal by Author**
>
> >  Can you provide a proof for equation 2?
>
> We have included a proof for Eq 2 (Eq 3 in the revised manuscript) in Appendix C (pasted below). The objective function for training the counterfactual generator minimizes the difference between $f\_\theta(\cdot|\overline{a})$ and the true counterfactual distribution $f\_{\overline{a}}$ with respect to a distributional difference $D_f(\cdot,\cdot)$ over all treatment combinations, $\overline{a}$. When the distance measure is the KL-divergence, this  equals maximizing the log-likelihood of the conditional distribution when data is sampled from the counterfactual distribution:
> $$\hat{\theta} = argmin\_{\theta \in \Theta} \mathbb{E}\_{\overline{a}} [{\rm KL}(f\_{\overline{a}}(\cdot) || f\_\theta(\cdot|\overline{a})) ]  $$
> $$= argmin\_{\theta \in \Theta} \mathbb{E}\_{\overline{a}} [\int \log \left(\frac{f\_{\overline{a}}(y)}{f\_\theta(y|\overline{a})}\right)f\_{\overline{a}}(y) dy]$$
> $$= argmax\_{\theta \in \Theta} \mathbb{E}\_{\overline{a}} [ \int \log \left(f\_\theta(y|\overline{a})\right)f\_{\overline{a}}(y) dy ] $$
> $$= argmax\_{\theta \in \Theta} \mathbb{E}\_{\overline{a}} [\mathbb{E}_{y \sim f\_{\overline{a}}} ~\log f\_\theta(\cdot|\overline{a}) ]$$
>
> > What would be the difference in objective if we were to define the distance in terms of Maximum Mean Discrepancy (MMD) or Wasserstein distance?
>
> We specifically study KL-divergence because it is well formulated for majority of generative models, including CVAE and many others, and has been focused in related work on counterfactual density estimation (Kennedy 2021, Melnychuk, 2023). Replacing KL with (kernel) MMD or Wasserstein-1 distance may lead to technical difficulties, due to the min-max formulation in MMD and Wasserstein distance. In particular, we have the following unified objective function for kernel MMD and Wasserstein-1 distance:
> $$
>     \hat{\theta} = argmin\_{\theta \in \Theta}~ \mathbb{E}\_{\overline{a}} [\sup\_{\phi \in \Phi} | \mathbb{E}\_{y \sim f\_\theta(\cdot | \overline{a})}[\phi(y)] -\mathbb{E}\_{y \sim f\_{\overline{a}}(\cdot)}[\phi(y)] \| \].
> $$
> Here $\Phi$ is a class of functions. For kernel MMD, $\Phi$ is class of kernel mappings and for Wasserstein-1 distance, $\Phi$ consists of all $1$-Lipschitz functions. As can be seen, the objective function now involves an inner maximization operation, which often leads to numerical instability. This phenomenon is well recognized in generative adversarial networks' training (Mescheder et al., 2018; Goodfellow et al., 2020). Therefore, we would like to emphasize the computational efficiency brought by Proposition 1 for the KL case. This proposition allows us to bypass the direct estimation of $f_{\overline{a}}$, significantly enhancing computational scalability in high-dimensional cases.
>
> > Considering your framework, it appears straightforward to expand it to the individual outcome level. What led to the decision to overlook that possibility?
>
> We appreciate the reviewer's reference to Individual Treatment Effects (ITE). We would like to clarify that our approach to estimating the counterfactual distribution across a population, rather than individual-focused estimates (ITE), is driven by the practicality of our research's applications, such as in public health policy. Implementing policies on an individual basis can be costly and sometimes unfeasible. By focusing on the population level (or a sub-population), we capture individual differences, which is essential for effective policy-making. In contrast, the variation within individual counterfactuals often stems from observational noise and might need substantial data for precise estimation. Consider, for instance, the COVID-19 data: our objective is to analyze the counterfactual outcome of a state-wide policy (mask mandate) and provide policymakers with a range of possible counterfactual outcomes that reflect variability in county-level confirmed cases. In such contexts, policymakers might be less concerned with the estimation noise at the individual level and more focused on the broader variability across different individuals.

---

> ### Author Response · Authors · 2023-11-19
> **Rebuttal by Author**
>
> (Continued from the last response)
>
>  In the meantime, a related extension of our work might be to infer conditional counterfactual outcomes (related to the conditional average treatment effect, CATE). This corresponds to looking at the distribution of the outcome under a specific subpopulation. Since our covariates are assumed to be time-varying, a common approach is to introduce a set of static baseline covariates, $V$ (Robins et al, 1999). The baseline covariate $V$ denotes the static feature (such as a patient's gender or age) that will influence both the time-varying covariates $X$ and the outcome $Y$. We can then draw counterfactual samples from a specific sub-population by conditioning on the values of the $V$. In this framework, we observe ($Y_t^i$, $\overline{A}_t^i$, $\overline{X}_t^i$, $V^i$), where $V \in \mathbb{R}^{
>     \nu}$ is a static baseline variable that varies by individual.  Accordingly, the generator will have an additional input:
> $ g\_{\theta}(z, \overline{a},v): \mathbb{R}^r \times \mathcal{A}^d \times \mathbb{R}^{\nu} \rightarrow \mathcal{Y}$ , and the IPTW weights will become $ w\_\phi(\overline{a},\overline{x},v) = \frac{1}{\prod\_{\tau=t-d+1}^{t} f\_\phi(a\_{\tau}|\overline{a}\_{\tau-1},\overline{x}\_{\tau},v)}$.   Correspondingly, the objective in Proposition 1 will become $\mathbb{E}\_{v} ~\mathbb{E}\_{\overline{a}} ~\left[\mathbb{E}\_{y \sim f\_{\overline{a},v}} ~\log f\_\theta(y|\overline{a},v)\right] \approx \frac{1}{N}\sum\_{(y,\overline{a},\overline{x},v) \in \mathcal{D}}w\_\phi(\overline{a},\overline{x},v) \log f\_\theta(y|\overline{a},v).$
>
>  We have conducted additional experiments using the fully synthetic dataset, where the baseline variable, $V$, was divided into two groups. The $V$ was uniformly drawn from $[-1,0]$ in the first group and from $[0,1]$ in the second group. We then looked at the performance of the generative samples corresponding to two sub-groups. We included the results in **Appendix I** (Table 4, Fig.11, and Fig.12) of the revised manuscript.  Our method (MSCVAE) still consistently outperforms other baseline methods in the simplified results below (Wasserstein distance, smaller the better):
>
> - $V\in[-1,0]$
>
> |         | d=1  | d=3  |  d=5  |
> | ----------- | ----------- | ----------- |----------- |
> | MSM+NN      |   0.408    |  0.449   |  0.368 |
> | KDE               |      0.201 |  0.562 |  0.564  |
> | Plugin+KDE   |        0.117    |      0.134      |   0.196      |
> | G-Net             |        0.431        |      0.823       |   0.843    |
> | CVAE             |       0.240          |     0.571       |   0.585   |
> | MSCVAE        |        0.051         |      0.083      |    0.186   |
>
> - $V\in[0,1]$
>
> |         | d=1  | d=3  |  d=5  |
> | ----------- | ----------- | ----------- |----------- |
> | MSM+NN      |   0.407    |  0.466  |  0.388 |
> | KDE               |      0.222 |  0.548 |  0.561  |
> | Plugin+KDE   |        0.121   |      0.109      |   0.182      |
> | G-Net             |        0.452       |      0.739       |   0.768    |
> | CVAE             |       0.258          |     0.565      |   0.591   |
> | MSCVAE        |        0.047         |      0.068      |    0.172   |

---

> > ### Author Response · Authors · 2023-11-23
> > **Follow-up**
> >
> > Dear reviewer,
> >
> > We sincerely hope that our responses have addressed your concerns. Please let us know if you have any additional questions or comments. We would be more than happy to follow up with additional details. Thanks again for taking the time to review our work.

---

### Official Review · Reviewer_iR6m · 2023-11-01

**Soundness:** 2 fair
**Presentation:** 2 fair
**Contribution:** 2 fair
**Rating:** 3
**Confidence:** 4

**Summary:**

The paper introduces a framework that can be used to simulate counterfactual outcomes in temporal experiments. The proposed method is a combination of conditional variational autoencoder and inverse probability weighting.

**Strengths:**

The proposed framework is simple, easy to use, and accessible. Judging from the experiments, the performance of the proposed method seems to be good.

**Weaknesses:**

- Proposition 1 doesn’t make sense. How could $\bar{a}$, a fixed value, be drawn from $\mathcal{D}$? Why is it a sum instead of an average? Wouldn’t the RHS of (4) be the same for all $\bar{a}$ while the LHS is supposed to be different? And why is the index in (5) from $t-d$ instead of from $t-d+1$? In proof of proposition 1, where does the expectation over $\bar{a}$ come from? I would be concerned if the authors actually used this formula in their experiments.

- I don’t seem to understand the comment in Remark 1 that doubly robust methods are less robust to model misspecification than IPW methods, and I cannot find relevant discussions in Appendix D as claimed. I'm curious about why the authors would think so.

- One pivotal assumption is that d, the length of history dependence, is finite and known, in which case the IPW methods in a temporal experiment are only a trivial extension to IPW methods in a static experiment. Also, as d gets large, the variance of those IPW-style methods can easily blow up.

- The only theoretical guarantee provided in the paper is that the weighted log likelihood is unbiased. In that sense, this is closer to treatment effect estimations where the estimand of interest is a single value, and it is very different from density estimations. Kennedy et al. (2023) (and some of the other papers mentioned by the authors) require stronger conditions simply because their goals are a lot harder. Since this paper is purely applied, I don’t see how they are comparable.

- Related to the last point, I fail to see how the absence of a unified theory for doubly robust density approximation in longitudinal settings serves as a reason for not using DR estimators, given that the authors never estimated the density. When the goal is not density estimation, there have been a plethora of studies on variations of IPW and AIPW estimators in longitudinal settings, especially from the dynamic treatment regimes and reinforcement learning literature.

- I find the notations occasionally confusing, and the authors are somewhat vague regarding the assumptions.

**Questions:**

See weaknesses.

---

> ### Author Response · Authors · 2023-11-19
> **Rebuttal by Authors**
>
> We thank the reviewer for the comments. We appreciate your input and regret any confusion that may have arisen if our methodology wasn’t adequately elucidated. We would like to clarify that our generative framework aims to draw high-quality counterfactual outcomes, which can be viewed as an implicit distribution estimator, bypassing the need for direct density estimation. We want to emphasize that this objective differs drastically from the majority of the existing literature, which only focus on estimating the mean of the counterfactual distribution. Please see our point-by-point responses below.
>
> > How could $\overline{a}$, a fixed value, be drawn from $\mathcal{D}$? Wouldn’t the RHS of (4) be the same for all $\overline{a}$
>  while the LHS is supposed to be different?
>
> We thank the reviewer for pointing it out.  Our generator approximates the underlying counterfactual for any $\overline{a}$,, so there should be an expectation over $\overline{a}$, for the objective function (Equation 2):
>     $$\hat{\theta}
>     = argmin_{\theta \in \Theta}~\mathbb{E}_{\overline{a}} ~\left[D_f(f\_{\overline{a}}, f\_\theta(\cdot|\overline{a})) \right],$$
>
> where $D$ represents the distributional distance between $f\_{\overline{a}}$ (true counterfactual distribution) and $f\_\theta$ (the proxy conditional distribution represented by our proposed counterfactual generator $g\_\theta$), such as KL divergence. The resulting weighted loss function in Proposition 1 becomes:
>
> $$
> \mathbb{E}\_{\overline{a}} [\mathbb{E}\_{y \sim f\_{\overline{a}}} ~\log f\_\theta(y|\overline{a})] \approx
>         \frac{1}{N}\sum\_{(y,\overline{a},\overline{x}) \in \mathcal{D}}w\_\phi(\overline{a},\overline{x}) \log f\_\theta(y|\overline{a})
> $$
>
> > Why is it a sum instead of an average?
>
>  We note that, in Proposition 1, we used the sum for simplicity, which is numerically equivalent to the average in practice as we fixed the batch size in our training.
>
> > And why is the index in (5) from $t-d$ instead of from $t-d-1$?
>
>   The index in (5) should be from $t-d+1$ instead of $t-d$, which is a typo.  We have fixed in the revised paper.
>
> >  Where does the expectation over $\overline{a}$ come from?
>
> The$\overline{a}$in the proof for Proposition 1 is due to that we were taking an expectation over $\overline{a}$  (as noted above). We have modified our notations in the revised manuscript.
>
> > I don’t seem to understand the comment in Remark 1 ...
>
> We would like to clarify that we did not suggest that the DR method lacks robustness under model misspecification than IPTW methods. In Remark 1, we intended to convey that it was challenging to extend our IPTW framework to a DR method. This is because in doing so, we need to augment the IPTW estimator with an outcome model for approximating the counterfactual distribution.  Such an extension would require accurately approximating the conditional outcome distribution $f(Y_t|\overline{X}_t,\overline{A}_t)$ and the conditional covariate distribution $f(X_t|\overline{X}\_{t-1},\overline{A}\_{t-1})$. This becomes particularly challenging when $Y$ is high-dimensional, a difficulty that is further exacerbated in the time-varying case with the need to accurately approximate $f(X_t|\overline{X}\_{t-1},\overline{A}\_{t-1})$. The G-Net is the only outcome-based method known to us that addresses such problem, and we have therefore included it as a baseline in our study. Our analysis revealed that G-Net's performance is suboptimal due to the aforementioned challenges, leading us to opt for the IPTW-only framework. We have clarified these points in the Appendix E of the revised paper.
>
> Furthermore, we would like to emphasize our main contribution is to use a generative model to draw high-quality counterfactual samples that follow the underlying counterfactual distribution. We opted for IPTW due to its seamless integration with our generative framework (the weighted loss in Proposition 1) and suitability for the high-d time-varying density approximation (it's easier to estimate the binary propensity model).
>
> > One pivotal assumption is that d, the length of history dependence, is finite and known...
>
>
>    In line with established practices in longitudinal causal inference, as referenced in works like (Robins et al, 1994; Lim et al, 2018; Bica et al, 2020; Li et al, 2021; Bica et al 2021), we selected $d$ based on our pre-existing understanding of the data generation process. This approach is validated by our numerical findings, proving to be an effective method. For instance, in our COVID-19 studies, we set $d=3$ following insights from epidemiological research, which indicates that a mask mandate typically requires two to three weeks to become effective in a population.

---

> > ### Author Response · Authors · 2023-11-19
> > **Rebuttal by Authors**
> >
> > > Also, as d gets large, the variance of those IPW-style methods can easily blow up.
> >
> > We agree that IPTW can be unstable and might lead to less reliable results in certain contexts.  Nevertheless, it's important to highlight that integrating IPTW with our novel generative framework represents a significant and novel methodological advancement. This combination is particularly effective in capturing complex, high-dimensional patterns in counterfactual outcomes, an area our study uniquely addresses to the best of our knowledge. To address the stability concerns in a practical context, our experiments incorporated established stabilization strategies, such as quantile truncation and standardization, as extensively discussed in the literature (Xiao et al, 2010; Chesnaye et al, 2022). The results from our experiments also show that incorporating these techniques into the design of our marginal structure model greatly results in promising and dependable numerical outcomes.
> >
> > > The only theoretical guarantee provided in the paper is that the weighted log likelihood is unbiased...
> >
> > We would like to clarify a potential misunderstanding: our technique should be seen as an implicit counterfactual distribution estimator. This is in contrast to standard density estimation methods and extends beyond merely estimating the mean, a subject thoroughly explored in existing research. A key strength of our generative modeling approach is its efficiency in sampling from the estimated counterfactual distribution. The generated samples enable a comprehensive understanding of the time-varying treatment effect and improve decision-making support for policymakers. Specifically, our method involves feeding random noise $z$ into our conditional generator to draw realistic random samples, as detailed in Algorithm 1. With high-dimensional outcome, our approach is far more feasible than what would be required with a traditional density estimation method, where estimating and sampling the density can become computationally prohibitive.
> > To elucidate how our method estimates the counterfactual distribution, we highlight the use of a conditional generator, $ g_\theta(z, \overline{a})$, to approximate this distribution. This approach is further detailed in Equations 2 of our revised manuscript, where the primary goal is to minimize distributional discrepancies between $f\_\theta(\cdot| \overline{a})$, the proxy conditional distribution  (approximated by $ g\_\theta(z,  \overline{a})$), and $f_{ \overline{a}}$, the actual counterfactual distribution:
> > $$\hat{\theta}
> >     = argmin_{\theta \in \Theta}~\mathbb{E}_{\overline{a}} ~\left[D_f(f\_{\overline{a}}, f\_\theta(\cdot|\overline{a})) \right].$$
> >
> > While similar objectives are explored in the works of Kennedy et al. (2021, 2023) and Melnychuk (2023), our method diverges in that we employ a conditional generator rather than a density estimator. In fact, this generative framework has been widely applied to approximate high dimensional distributions in fields such as computer vision. For example, generative models such as the Variational Auto-Encoder (VAE) and Generative Adversarial Networks (GAN) are capable of producing a variety of images that reflect the underlying distribution of the training dataset, without explicitly estimating the density of the high-dimensional image distribution.
> >
> > > Related to the last point... when the goal is not density estimation...
> >
> > We agree with the reviewer that if our goal was to estimate a single value such as the mean, there have been a plethora of studies in longitudinal settings. However, as stated in the previous point, our method indeed requires the generated samples to match the counterfactual distribution by using the generative model to approximate such a distribution. Therefore, the absence of a unified theory for doubly robust density approximation in longitudinal settings serves as a major reason for not using DR estimators. As previously mentioned, we explored the outcome-based method, G-Net, which can be potentially combined with IPTW for a DR framework. However, we encountered significant challenges in effectively combining it with our IPTW method due to the sub-optimal empirical performance of G-Net, a result of the methodological challenges discussed above. We remain open and enthusiastic about the possibility of adopting the DR approach, especially if future insights emerge regarding the development of an outcome-based method capable of accurately approximating high-dimensional, time-varying counterfactual distributions.
> >
> > > Notations occasionally confusing... vague regarding the assumptions
> >
> > We thank the reviewer for pointing this out. We would like to mention that the assumptions of our methodology are detailed in Appendix A. Furthermore, we have updated the manuscript to enhance the clarity of the notations. If there are still any notations that the reviewer finds unclear, we would highly value specific suggestions for further improvement.

---

> > > ### Comment · Reviewer_iR6m · 2023-11-22
> > >
> > > Thanks for answering my questions and addressing some of my comments. A few things:
> > >
> > > - Proposition 1 now looks a lot better with all those typos fixed. Nevertheless, proof of proposition 1 still seems to be wrong. The authors somehow didn't condition on $\bar a$ so there are steps where they took expectation over $\bar a$ twice. Also, I found the proof incredibly hard to read as a conventional unbiasedness proof of an IPW estimator, for that the authors basically used $f(\cdot)$ to denote everything.
> > >
> > > - I never said the authors didn't attempt to estimate the distribution. Although the authors aim to estimate the distribution, they only show that the log-likelihood objective, which appears to me to be a single value, is identifiable and can be estimated unbiasedly. Thus, if the authors do believe unbiased estimation of the log-likelihood suffices for estimating distribution, then the DR estimator, being unbiased as well, would also be applicable, and that is why I think the absence of a unified theory for doubly robust density approximation doesn't seem to be a valid reason. However, I'm not suggesting that the authors must use a DR estimator or provide theoretical guarantees. It is just my opinion that it might be easier to simply acknowledge limitations rather than make claims on methods that might not be fully understood.

---

> > > > ### Author Response · Authors · 2023-11-23
> > > > **Responses to Comment by Reviewer iR6m**
> > > >
> > > > We appreciate the reviewer for your comments. Here are our responses.
> > > >
> > > > > Proposition 1 now looks a lot better with all those typos fixed. Nevertheless, proof of proposition 1 still seems to be wrong...
> > > >
> > > > We do acknowledge the possible confusion throughout the proof. Therefore, we have thoroughly updated the proof of Proposition 1 (Appendix D, page 16 of the latest version of the manuscript).  We generally followed the definition and derivation in the slides and chapters of (Robins and Hernán, 2009; Robins, 2018). For clarity, we have also added precise definitions of the notations before the proof and also in-line comments hopefully to make the proof easier to read.
> > > >
> > > >    We would also like to clarify that in the original proof of Proposition 1, the inner integration over $\overline{A}$ is due to the definition of the counterfactual density, which involves integrating the IPW-weighted density of the observation over $\overline{A}$  and $\overline{X}$ and only picking $\overline{A}=\overline{a}$. Therefore this inner integration is not evaluated over different $\overline{a}$. By presenting the proof in a more succinct and clear format in the updated manuscript, we hope such confusion is addressed.
> > > >
> > > > >  I never said the authors didn't attempt to estimate the distribution. Although the authors aim to estimate the distribution, they only ...
> > > >
> > > > We thank the reviewer for the clarification. We agree that in theory, we can extend our IPW framework to the DR setting, and the main reason why we did not opt for DR is due to the **practical** challenges in estimating the outcome model under the high-d outcome settings (as explained in our previous responses). We have further updated the manuscript (Remark 1 and Appendix E) to make it clear that our framework can be potentially extended to the DR framework, and stated the practical reasons why we did not.

---

> > > > > ### Comment · Reviewer_iR6m · 2023-11-23
> > > > >
> > > > > Thanks. If the authors hope to use $\bar{A}$ and $\bar{a}$ to distinguish between a random variable and a constant, then how could you take expectation over a constant?

---

### Author Response · Authors · 2023-11-19
**Response to AC and all reviewers**

We thank all reviewers for their valuable comments and suggestions on our manuscript. We're grateful for the recognition of the significance our research topic, the flexibility of generative models for estimating high-dimensional counterfactual distributions, and the robust performance of our methods, particularly on real datasets. We have updated the manuscript (with blue fonts) to address the questions from each reviewer. Below are the main points we have addressed:

- We have updated the mathematical notations for better clarity, especially in Equation 2 and Proposition 1. Our objective is to minimize the distributional distance between a proxy conditional distribution (represented by our conditional generator) and the counterfactual distribution under **all** treatment combinations:  $\hat{\theta} = argmin\_{\theta \in \Theta}~\mathbb{E}\_{\overline{a}} [D_f(f\_{\overline{a}}, f\_\theta(\cdot|\overline{a})) ]$ , where $D$ represents the distributional distance between $f_{\overline{a}}$ and $f_\theta$, such as the KL divergence, and $\theta$ represents the parameters of our conditional generator. This has also clarified the reviewers' questions on the proof of Proposition 1 and whether $\overline{a}$ is fixed.


-  We have pointed out that our main contribution is to use a conditional generative model to produce high-quality counterfactual outcomes. The generated samples enable the comprehensive understanding of the time-varying treatment effect and improves decision-making support for policymakers. It acts as an implicit distribution estimator, bypassing the need for direct density estimation. This method significantly deviates from most existing research, which primarily concentrates on mean estimation. Moreover, other attempts to estimate counterfactual density are not easily adaptable to our scenario due to the challenges posed by high-dimensionality in conventional density estimation.


- We clarify that our approach is for estimating the counterfactual distribution across a population, rather than individual-focused estimates (ITE). This is driven by the practicality of our research's applications, such as in public health policy. Implementing policies on an individual basis can be costly and sometimes unfeasible. By focusing on the population level, we capture individual differences, which is essential for effective policy-making. In contrast, the variation within individual counterfactuals often stems from observational noise and might need substantial data for precise estimation. We have also added additional experiments to show our method can be extended to draw  counterfactual samples under a sub-population (related to CATE).

---

> ### Author Response · Authors · 2023-11-22
> **Follow-up**
>
> Dear reviewers,
>
>
> We hope that our responses have addressed your concerns. Please let us know if you have any additional questions or comments. We would be more than happy to follow up with additional details. Thanks much for taking the time to review our work.

---

### Meta-Review · Area_Chair_Qioq · 2023-12-12

**Metareview:**

This paper develops a technique for estimating counterfactual distributions using a generative models. The approach is very simple in that it uses inverse probability weighting to construct a target from the observed data for the generative model. The generative model used is a conditional variational autoencoders, though the authors note that other generative models can be used. There was variance among the reviewers with one reviewer initially positive and another reviewer moving to a weak accept. Even after update, the reviews on average are more negative. I took a the paper myself and my enthusiasm was muted in the paper looks like combines two things together (inverse probability weighting for a target and generative modeling) in a very straightforward manner and the experiments are limited. As an example, the history of dependence is quite short. This paper has potential, but it would need to clarify the contribution and some of the loose surrounding text (such as the initial text on double robustness) and have a stronger set of experiments.

**Justification For Why Not Higher Score:**

The paper had mixed feelings. There was no overwhelmingly positive support. From my look at the paper, it seemed very straightforward in way that made it hard to tell where the added value was relative to what is already known.

**Justification For Why Not Lower Score:**

NA

---

### Decision · Program_Chairs · 2024-01-16

Reject